# Supraglacial streamflow and meteorological drivers from southwest Greenland

**Rohi Muthyala**[1], **Åsa K. Rennermalm**[1], **Sasha Z. Leidman**[1], **Matthew G. Cooper** `TS1`[2,3], **Sarah W. Cooley**[4], **Laurence C. Smith**[5,6], **and Dirk van As**[7]

[1]Department of Geography, Rutgers, The State University of New Jersey, New Brunswick, NJ 08901, USA

[2]Department of Geography, University of California, Los Angeles, Los Angeles, CA 90095, USA

[3]Atmospheric Sciences and Global Change Division, Pacific Northwest National Laboratory, Richland, WA 99354, USA

[4]Department of Geography, University of Oregon, Eugene, OR 97403, USA

[5]Institute at Brown for Environment and Society, Brown University, Providence, RI 02912, USA

[6]Department of Earth, Environmental, and Planetary Sciences, Brown University, Providence, RI 02912, USA

[7]Geological Survey of Denmark and Greenland, Øster Voldgade 10, 1350 Copenhagen, Denmark

**Correspondence:** Rohi Muthyala (rohi.muthyala.91@gmail.com)

**Abstract.** `TS2` Greenland ice sheet surface runoff is drained through supraglacial stream networks. This evacuation influences surface mass balance as well as ice dynamics. However, in situ `CE1` observations of meltwater discharge through these stream networks are rare. In this study, we present 46 discrete discharge measurements and continuous water level measurements for 62 d spanning the majority of of the melt season (13 June to 13 August) in 2016 for a 0.6 km$^2$ supraglacial stream catchment in southwest Greenland. The result is an unprecedentedly long record of supraglacial discharge that captures both diurnal variability and changes over the melt season. A comparison of surface energy fluxes to stream discharge reveals shortwave radiation as the primary driver of melting. However, during high-melt episodes, the contribution of shortwave radiation to melt energy is reduced by $\sim 40\%$ (from 1.13 to 0.73 proportion). Instead, the relative contribution of longwave radiation, sensible heat fluxes, and latent heat fluxes to overall melt increases by $\sim 24\%$, 6%, and 10% (proportion increased from $-0.32$ to $-0.08$, 0.28 to 0.34, and $-0.04$ to 0.06) respectively. Our data also identify that the timing of daily maximum discharge during clear-sky days shifts from 16:00 local time (i.e., 2 h 45 min after solar noon) in late June to 14:00 in late July and then rapidly returns to 16:00 in early August. The change in the timing of daily maximum discharge could be attributed to the expansion and contraction of the stream network, caused by skin temperatures that likely fell below freezing at night. The abrupt shift, in early August, in the timing of daily maximum discharge coincides with a drop in air temperature, `CE2` a drop in the amount of water temporarily stored in weathering crust, and a decreasing covariance between stream velocity and discharge. Further work is needed to investigate if these results can be transferable to larger catchments and uncover if rapid shifts in the timing of peak discharge are widespread across Greenland supraglacial streams and thus `CE3` have an impact on meltwater delivery to the subglacial system and ice dynamics.

## 1 Introduction

Mass loss from the Greenland ice sheet increased 6-fold from the 1980s to the 2010s ($286 \pm 20$ Gt yr$^{-1}$) (Mouginot et al., 2019). This mass loss was dominated by enhanced surface melting and runoff (van den Broeke et al., 2016). The increase in runoff raised Greenland's contribution to global sea-level rise from less than 5% in 1993 to more than 25% in 2014 (Chen et al., 2017). Increased surface melting also influences ice sheet basal properties (Das et al., 2008; Colgan et al., 2011; Flowers, 2018) and ice dynamics (van de Wal et al., 2008; Shepherd et al., 2009; Schoof, 2010; Hoffmann et al., 2011; Hewitt, 2013; Andrews et al., 2014). Though

the net effect of meltwater runoff on basal pressures and ice velocities remains unclear, recent studies show that, in the lower-ablation regions, increase in surface runoff results in a decrease in ice velocities (Sundal et al., 2011; Tedstone et al., 2015; Davison et al., 2019). For example, a 50 % increase in surface melting during 2007–2014, compared to 1985–1994, resulted in 12 % slower ice flow in the lower-ablation region (within ∼ 50 km of the margin) of west Greenland (Tedstone et al., 2015). Surface melting feeds numerous supraglacial stream/river networks that develop on the surface of the Greenland ice sheet ablation zone every melt season (Smith et al., 2015, 2017; Yang and Smith, 2013, 2016; Pitcher and Smith, 2019). These networks transport runoff sourced from melting ice, snow, and/or slush within the stream catchment (Holmes, 1955; Karlstrom et al., 2014) and often terminate in moulins, wherein meltwater moves within and beneath the ice sheet before emerging in proglacial rivers, lakes, fjords, and the ocean (Chu, 2014; Rennermalm et al., 2013). Despite the importance of Greenland's surface runoff to ice sheet dynamics and sea-level rise, only a handful of studies use in situ supraglacial stream discharge to characterize current conditions (Holmes, 1955; Chandler et al., 2013; McGrath et al., 2011; Gleason et al., 2016; Smith et al., 2015, 2017, 2021; Chandler et al., 2021), and these studies are limited to short periods except for in Wadham et al. (2016), who recorded a 50 d period of supraglacial discharge as a part of their study on the export of nitrogen from the Greenland ice sheet.

Supraglacial stream discharge varies in concert with surface melting, with low flow at the beginning and end of the melt season and higher flow in the middle of the melt season (Holmes, 1955). Supraglacial discharge also shows a pronounced diurnal variation (McGrath et al., 2011; Wadham et al., 2016; Smith et al., 2017, 2021; Yang et al., 2018). While daily maximum discharge varies with catchment size and day of the season, discharge decreases when melt energy drops off at night (Marston, 1983; Mernild et al., 2006; McGrath et al., 2011; Yang et al., 2018). Diurnal variability and timing of meltwater delivery to the subglacial drainage system have been shown to influence ice sheet velocities in several studies (Bartholomew et al., 2012; Sole et al., 2013; Andrews et al., 2014; Smith et al., 2021), with up to 65 % increase in ice velocity in the lower-ablation area (Sole et al., 2013). Short-term speedups occur in the lower-ablation regions of southwest Greenland (Shepherd et al., 2009), with an increase in ice velocities by up to 300 %–400 % (compared to pre-melt speeds) that lasts for a few days to a week in response to the variations in surface runoff supply (Sole et al., 2013). However, no observational studies have documented the diurnal variability in and timing of Greenland ice sheet supraglacial flow throughout an entire melt season.

The routing of meltwater through supraglacial stream networks as well as non-channelized surfaces delays the timing of peak discharge at the moulin relative to the timing of peak surface melt (Karlstrom et al., 2014; Yang et al., 2018). This peak time lag primarily depends on the size of the catchment and meltwater routing time (Holmes, 1955; Mernild et al., 2006; McGrath et al., 2011; Smith et al., 2017). Larger catchments imply a longer stream network and thus a larger time lag between peak surface melt and peak discharge compared to smaller catchments with similar surface melt intensity. Additionally, when a supraglacial stream network grows and shrinks throughout the melt season, the magnitude of peak moulin discharge decreases and increases respectively (Yang et al., 2018). For example, when the actively flowing network contracts (Lampkin and VanderBerg, 2014) and more water is transported via porous media flow, the peak time lag increases and the magnitude of peak moulin discharge can decrease CE4 by more than 50 % (Yang et al., 2018). Non-channelized meltwater predominantly flows or is temporarily stored in weathering crust (Cooper et al., 2018; Yang et al., 2018). Weathering crust is a degraded, porous surface layer of ice that retains meltwater temporarily, influencing the magnitude of peak discharge (0.14–0.18 m of specific meltwater storage within low-density ice; Cooper et al., 2018) and promoting subsurface flow (Karlstrom et al., 2014; Cooper et al., 2018). This may slow the transport of meltwater to supraglacial streams (Munro, 2011; Karlstrom et al., 2014; Cook et al., 2016; Yang et al., 2018; Gleason et al., 2021), delaying the time of peak discharge. In addition to the structure of weathering crust, the amount of meltwater stored is proportional to solar radiation as windy and overcast conditions with higher longwave radiation reduce the storage capacity of weathering crust (Takeuchi, 2000). However, the evolution of the timing of peak discharge and storage of meltwater in weathering crust through the melt season has never been reported in previous studies due to the short span of in situ data available.

In Greenland's ablation zone, seasonal and interannual variability in meltwater production is primarily driven by the variability in shortwave radiation absorption (van den Broeke et al., 2011; Ryan et al., 2019). Secondary melt drivers are turbulent fluxes of sensible and latent heat, particularly in the lower-ablation zone of southwest Greenland (van den Broeke et al., 2011; Fausto et al., 2016). Lesser drivers include anomalously moist and warm air masses advected over the ice sheet by atmospheric rivers (Mattingly et al., 2018) and clouds with contrasting feedback to surface melt (Bennartz et al., 2013). While a few studies report that an increase in cloud cover enhances downward longwave radiation and hence melt (Van Tricht et al., 2016; Gallagher et al., 2020), others show that it also limits shortwave radiation, thus decreasing summer melt in ablation areas (Hofer et al., 2017; Izeboud et al., 2020). It remains unclear which of these radiation components dominantly drives surface melt during cloudy conditions over the Greenland ice sheet. Numerous studies examine the linkages between surface energy balance, surface melting, and runoff using regional climate models (e.g., Fettweis et al., 2017; Noël et al., 2018) and automatic weather station data (van As et al., 2012). In contrast to model-simulated surface melt and runoff, observed

meltwater delivery to moulins (i.e., supraglacial discharge) is affected by processes influencing surface flow and storage of water in weathering crust and is CE5 thus a better representation of meltwater that actually leaves the ice sheet surface. However, relatively few studies compare surface energy fluxes with in situ observations of supraglacial stream discharge in Greenland (Smith et al., 2017) and no study compares their contribution to in situ stream discharge throughout the melt season.

Understanding supraglacial stream channel geometry is critical for determining the routing speed, potential sediment cover, and albedo of the supraglacial stream channel (Karlstrom and Yang, 2016; Leidman et al., 2021). Smith et al. (2015) and Gleason et al. (2016) used at-a-station hydraulic geometry theory to calculate how channel width, depth, and velocity co-vary nonlinearly with discharge for a fixed stream cross-section. This theory provides a set of equations with parameters that can be generalized to estimate discharge in ungauged rivers (Smith et al., 1996, 2015; Ashmore and Sauks, 2006; Andreadis et al., 2020; Leopold and Maddock, 1953; Ferguson, 1986; Gleason et al., 2015). Similarly, a generalization of supraglacial streams' hydraulic geometries would open possibilities for scaling and modeling discharge. For example, Smith et al. (2015) used field-calibrated hydraulic geometry to estimate instantaneous discharges in 523 moulins in southwest Greenland, yielding values ranging from 0.36 to 17.72 $m^3\,s^{-1}$ with a mean value of $3.15\,m^3\,s^{-1}$. In contrast, Gleason et al. (2016) and Smith et al. (2017) argued that unlike terrestrial systems, uniform hydraulic behavior cannot necessarily be expected from an ice substrate. Only a few studies have quantified hydraulic geometry of supraglacial streams, all using a relatively short data record (Knighton, 1981; Marston, 1983; Karlstrom et al., 2014) TS3.

In light of the current knowledge gaps in Greenland ice sheet supraglacial hydrology discussed above, this paper addresses the following questions: (i) how does supraglacial discharge vary over an entire melt season within a well-defined catchment? (ii) What drives these variations throughout the melt season? (iii) Do CE6 the timing and magnitude of daily peak discharge change throughout the season, and, if so, (iv) do the observed changes correspond to changing hydraulic geometry parameters in the supraglacial stream channel? We present a 62 d time series of supraglacial streamflow, in southwest Greenland, spanning most of the 2016 melt season (13 June–13 August 2016). The supraglacial stream drainage network was mapped using uncrewed aerial vehicle (UAV) imagery and field GPS observations. Surface energy fluxes were calculated or measured using meteorological observations from a nearby automatic weather station (KAN_L). From these data, we examine supraglacial stream discharge, diurnal variability, the daily maximum and uncertainties, and the contributions of meteorological drivers to this discharge throughout the melt season. We also estimate and compare hydraulic geometry parameters of the

supraglacial stream with previous studies. Finally, we explore how the time of daily maximum discharge evolves through the melt season. We conclude with a discussion of how change in the time of daily maximum discharge varies with air temperature, hydrologic geometry parameters, and the subsurface water level and with some recommendations for future research.

## 2  Study area

The study area is a $0.6\,km^2$ TS4 internally drained supraglacial catchment in southwest Greenland, hereafter called "660 catchment" (after "Point 660" where a gravel road from the town of Kangerlussuaq ends at the ice sheet margin). The catchment is located $\sim 1$–2 km upstream of the ice edge between two outlet glaciers, Isunnguata Sermia and Russell Glacier, and roughly 35 km east of Kangerlussuaq (Fig. 1). The stream network terminates in a moulin (location 67.1562° N, 50.0064° W in 2016). Elevations in the catchment span from 610 m near the gauging station to 660 m (above the WGS84 ellipsoid) at the catchment's highest point (Fig. 1).

The catchment surface consists of a rugged bare-ice landscape with small supraglacial ponds and an incised stream network (Fig. 2a). The bare-ice surface has an albedo of $0.57 \pm 0.04$ (Moustafa et al., 2015) and has a thin ($\sim 0.1$–0.3 m) surface layer of weathering crust comprising porous ice and cryoconite holes (Fig. 2c). These cryoconite holes are partially filled with water and accumulate cryoconite, consisting of dust, sediment, and biological matter (Takeuchi, 2000; Cooper et al., 2018). Cryoconite deposits are widespread in streams and ponds throughout the catchment (Leidman et al., 2021). The catchment is situated in a region where winter snow accumulation is relatively low and that experiences extensive melting from June through August so that little to no snow cover remains on the bare ice early in the melt season (Rennermalm et al., 2013; Ryan et al., 2019).

## 3  Data and methods

### 3.1  Supraglacial stream discharge

About 850 m upstream of the moulin, a gauging station for monitoring water level and discharge was installed at 67.1573° N, 49.9951° W. Stream water stage was measured using a setup of two Solinst pressure transducers: a Levelogger® in a perforated, weighted steel enclosure resting on the streambed tied to an embedded pole (Fig. 2b), which also supplies a fixed reference point throughout the season, and a Barologger® (Solinst, 2020) installed 25–30 m northeast of the gauging/discharge station. Stage is calculated after barometric pressure correction, yielding a continuous time series of stage measurements, recorded every 5 min from 13 June to 13 August 2016. This period covers most of the melt

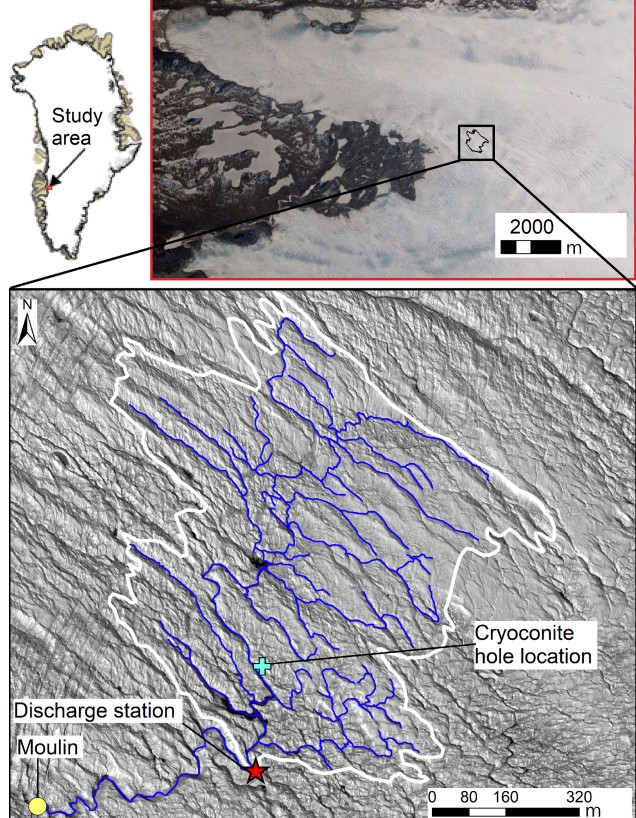

**Figure 1.** Map of the study site showing the supraglacial catchment boundary (white); streams of order 1, 2, and 3 (blue); and locations of the discharge station, cryoconite hole, and terminal moulin.

season in the study area and is hereafter referred to as the melt season.

Discharge was calculated with the velocity–area method using inputs of cross-sectional area and stream water veloc- ₅ ity (e.g., Herschy, 1993a). Stream velocity was measured at 60 % of the depth at each 0.2 m interval horizontally across the stream, with either a General Oceanics current meter or a Price Type AA current meter. Cross-sections of stream depth were measured at 0.2 m intervals across the 1.7–3.3 m wide ₁₀ stream. In total, 46 discrete observations of velocity and the cross-sectional area were made, including 27 measured every hour from 15:30 on 26 July 2016 to 17:30 on 27 July 2016 (local time) to capture the entire diurnal range. The remain- ing 19 observations were collected over the entire study pe- ₁₅ riod and sampled on average every 3–7 d between 12:00 and 17:00 local time. Though measurements were collected in the same location throughout the season, continuous ther- mal erosion of the bed resulted in small changes in cross- sectional geometry (Fig. 3a), consistent with previous stud- ₂₀ ies (Wadham et al., 2016; Smith et al., 2021) that measured long-term supraglacial stream discharge. The hourly mea- surements were collected with a fixed reference start point over a 26 h period between 26 and 27 July.

The discharge rating curve was generated with a best-fit power law (e.g., Herschy, 1993b): ₂₅

$$Q = q(H + \alpha)^{\beta}, \tag{1}$$

where $q$ and $\beta$ are constants estimated by fitting the curve to observations of discharge ($Q$) and water level, also called stage height ($H$), and $\alpha$ is the water level sensor offset from the stream bottom. In this study, the box with the ₃₀ Levelogger® was placed on the streambed ($\alpha = 0$).

Discharge was determined using a rating curve relating 46 discrete discharge measurements to continuous (5 min in- terval) observations of stream water stage. The rating curve ($Q = 3.925H^{2.44}$; coefficient of determination $R^2 = 0.94$; ₃₅ Fig. 4) was then used to generate continuous discharge val- ues from stage measurements recorded every 5 min through- out the season. These data were in turn averaged to yield a continuous record of hourly discharge data for 62 d, from 13 June to 13 August 2016 (Fig. 5a). ₄₀

Four uncertainty estimates were calculated as percentage uncertainties at the 95 % confidence interval (see Appendix A for more details): (1) uncertainty for the 46 discrete dis- charge measurements ($U_{me}$), (2) uncertainty due to the rating curve ($U_{RC}$), (3) uncertainty in daily mean discharge ($X_{dm}$), ₄₅ and (4) uncertainty due to the streambed incision into the ice over the melt season ($U_{in}$). $U_{me}$ was estimated at 10.8 %, and $U_{RC}$ was estimated at 17 % (Appendix A). The measure- ment uncertainties were encompassed in the envelope of un- certainty due to the rating curve ($Q \pm U_{RC}$) (Fig. 4). The aver- ₅₀ aging of hourly discharge to daily mean discharge generated an uncertainty of 25 % ($X_{dm}$) (Appendix A).

Finally, uncertainty due to streambed incision ($U_{in}$) was estimated using the cross-sectional profiles of the streambed (Fig. 3). Reconstructing hydrographs for supraglacial ₅₅ streams with high diurnal and melt season variations with a rating curve is typically unreliable (Smith et al., 2017, 2021; Pitcher and Smith, 2019). In terrestrial rivers, shifts in the rating curve are a reflection of either a datum adjustment or changes in the channel cross-section. Unlike terrestrial ₆₀ rivers, the bed under supraglacial streams is constantly melt- ing and incising into the ice, resulting in an ever-changing cross-sectional profile. This melting may or may not alter the geometry of the stream cross-section. To examine if our rating curve is robust despite channel cross-sectional profile ₆₅ changes, we compared coincident depth profiles and veloc- ity measurements (Fig. 3). The discrete discharge measure- ments over a cross-section are susceptible to both measure- ment and incision errors. However, assuming that negligible incision occurs over the 26 h period, uncertainty in hourly ₇₀ discharge measurements could be attributed to measurement errors alone. Therefore, by separating profiles collected over high flows through the season from profiles collected over the 26 h period of hourly measurements, measurement er- rors can be isolated from incision errors. While profiles col- ₇₅ lected over the season show a 0.037 m standard deviation (Fig. 3a), hourly profiles collected over the 26 h period show

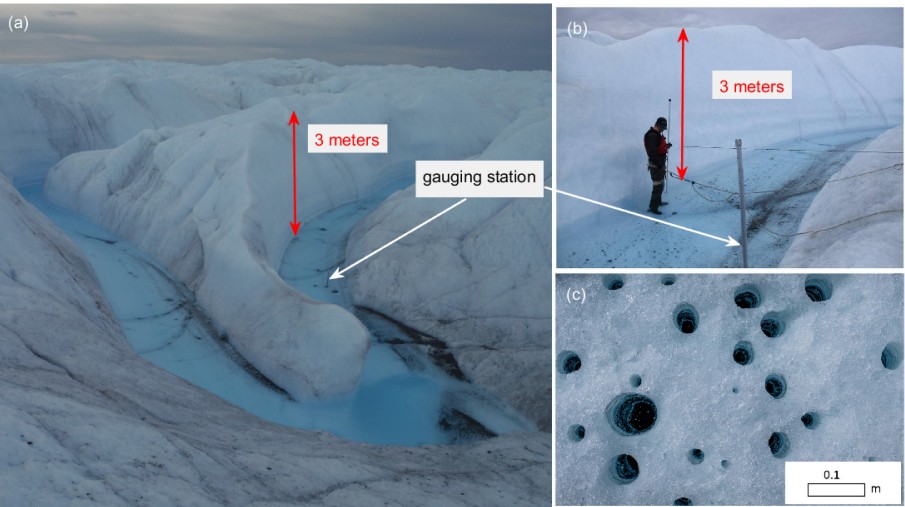

**Figure 2. (a)** Supraglacial stream main stem and gauging station location. **(b)** Close-up photo of the stream cross-section and the gauging station during discharge measurement. **(c)** Cryoconite holes on the bare-ice surface vary from 0.02–0.08 m in diameter and 0.1–0.3 m in depth (in this figure). These holes are partially water filled and contain cryoconite (biological matter, dust, and sediment) at the bottom.

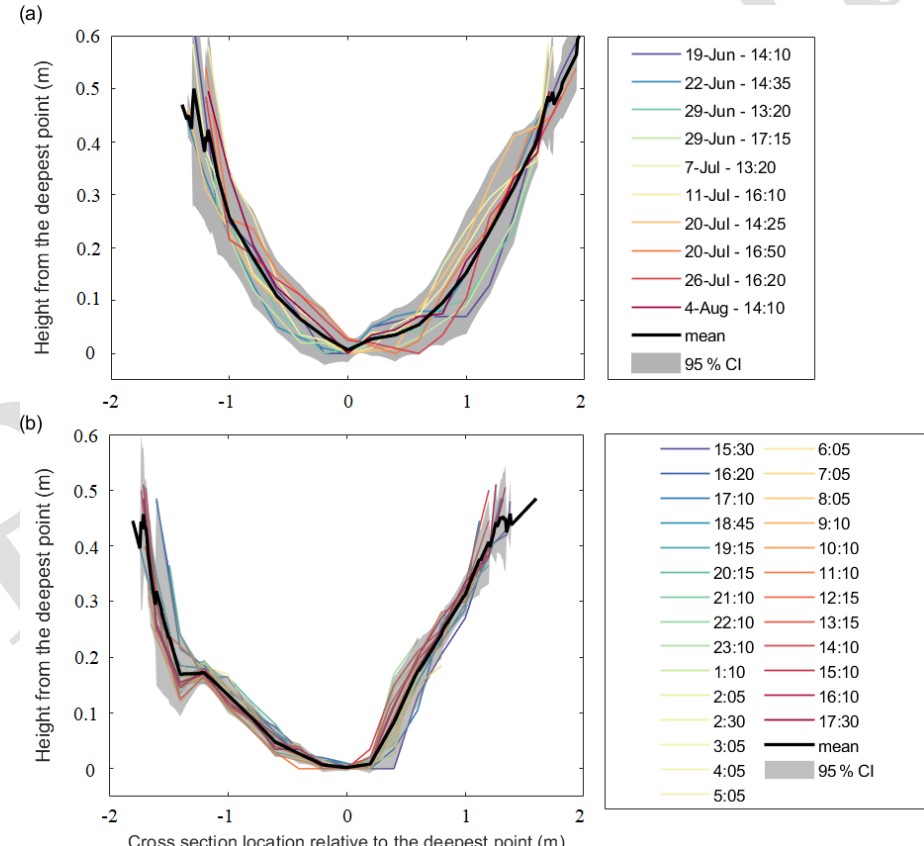

**Figure 3.** Stream cross-section depth profiles along the wetted perimeter: **(a)** daily cross-sections using measurements performed from 19 June through 8 August, with samples collected on average every 3–7 d, and **(b)** hourly cross-sections using measurements from 15:30 on 26 July to 17:30 on 27 July 2016. The horizontal axes in these depth profiles have been adjusted so that the zero points co-occur with the maximum depth in each sample (due to which the profiles do not align perfectly with each other in panels **a** and **b**). The mean profile is shown in a thick black line, and uncertainty (95 % CI) is shown in the grey-shaded area. All times correspond to local time.

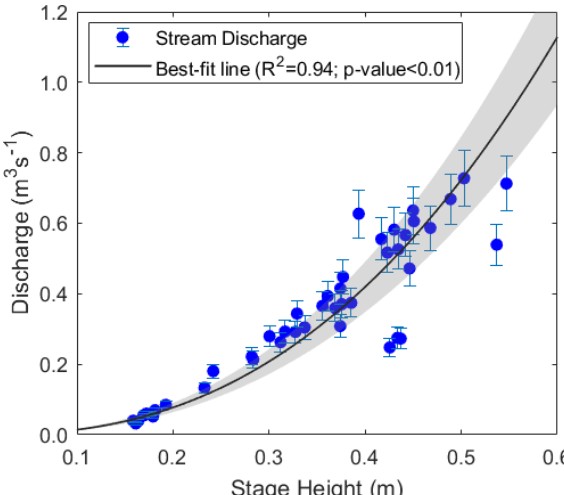

**Figure 4.** Rating curve (black line) determined from the best fit of the power law (Eq. 1) to 5 min continuous measurements of stage corresponding to discrete measurements of discharge (blue dots). Error bars show measurement error uncertainty ($U_{\text{me}}$). Rating curve uncertainty ($U_{\text{RC}}$) is shown in the grey-shaded area at the 95 % CI (see Appendix A for uncertainty calculations).

a 0.019 m standard deviation (Fig. 3b). The uncertainty in stream discharge (here we use the 95 % confidence interval) due to non-uniform streambed incision and depth measurement errors, $U_{\text{in}}$, is 10.9 % (of the average stream depth), and the depth measurement alone is 5.9 %. Despite these errors, the channel geometry incises uniformly through the season with all the cross-sections lying inside the uncertainty levels (Fig. 3a). Therefore, we conclude that our rating curve is sufficiently robust to estimate discharge.

## 3.2 Water level in weathering crust

Storage of meltwater in weathering crust is investigated by measuring the water level in a cryoconite hole using a Levelogger®, similar to the one used to measure water level in the stream (Solinst, 2020). The Levelogger® was placed at the bottom of the cryoconite hole, for a 3-week period from 24 July to 13 August, located in the 660 catchment close to the station where discharge measurements were collected. A Barologger® was also placed at the location for barometric pressure correction of the water level.

## 3.3 Calculation of hydraulic geometry parameters

At-a-station hydraulic geometry parameters were calculated to examine the relative importance of width, velocity, and depth in controlling discharge and to compare with other studies reporting hydraulic geometry data for supraglacial streams. The hydraulic geometry power-law equations are

(Leopold and Maddock, 1953)

$$w = aQ^b, \qquad (2)$$
$$d = cQ^f, \qquad (3)$$
$$v = kQ^m, \qquad (4)$$

where $w$, $d$, and $v$ are the stream width, depth, and velocity of the cross-section respectively. The exponents $b$, $f$, and $m$ represent the slopes of the power-law equations. The magnitude of the exponents represents the rates of change of each variable with respect to the independent variable, discharge $Q$. The coefficients $a$, $c$, and $k$ represent the $y$-axis intercepts. The law of conservation of mass implies that the product of coefficients ($a$, $c$, and $k$) and the sum of the exponents ($b$, $f$, and $m$) should equal 1 (Leopold and Maddock, 1953).

## 3.4 Automatic weather station observations in the 660 catchment

To identify the timing of daily maximum melt during clear-sky days, meteorological observations were obtained from the nearby automatic weather station (AWS), KAN_L (van As et al., 2011), maintained by the Geological Survey of Denmark and Greenland (GEUS) and located $\sim$7–8 km southeast of our study area at 670 m elevation. We also installed a shortwave pyranometer (HOBO S-LIB-M003) at a 2 m height $\sim$20–25 m from the gauging station, but these data were not used in this study since KAN_L offered better data continuity and measurements of other surface energy components.

## 3.5 Surface energy balance model

To examine surface energy drivers of supraglacial discharge, energy balance components were obtained from a surface energy balance model (described in van As, 2011). This model uses forcing data from in situ meteorological and radiative observations from KAN_L to calculate the surface energy balance components net shortwave radiation, net longwave radiation, sensible heat flux, latent heat flux, subsurface conductive heat flux (i.e., ground heat flux), and heat flux from rain. While incoming shortwave and longwave radiation are gathered from the AWS, turbulent heat fluxes are calculated from near-surface gradients of meteorological variables, air temperature, humidity, and wind speed, using Monin–Obukhov similarity theory (van As, 2011).

## 3.6 Catchment delineation

The catchment boundary and supraglacial stream network were manually digitized using two sources (Fig. 1). Firstly, we used WorldView-1 (WV1) panchromatic imagery (spatial resolution of 0.5 m) acquired on 16 August 2016 to manually digitize the stream network. We also collected 20 000 hand-held GPS points of the catchment boundary in the field by walking along the visually determined catchment divide on

15 August 2016. We did not observe a change in catchment size during the study period. We estimate the catchment area to be accurate within 5 % (0.03 km$^2$) given that it was manually identified in the field. However, the precise delineation of the catchment is not relevant to the outcome of the study.

## 4 Results

Our catchment has a dendritic drainage pattern (Strahler order 4, as determined from our manual digitization) and is internally drained, meaning that all surface meltwater is routed through streams, tributaries, and ponds to a terminal moulin (Fig. 1). Repeated visits to the study site during the melt season suggested that the majority of streams re-occupied existing channel networks between 2015–2019, resulting in similar channel depths and widths to those in 2016 CE7.

### 4.1 Hourly and daily variations in supraglacial discharge

Stream discharges between 13 June and 13 August vary strongly both diurnally and over the melt season (Fig. 5a). Hourly discharge fell as low as 0.002 m$^3$ s$^{-1}$ at night, and daily peaks exceeded 0.3 m$^3$ s$^{-1}$ on most days. Three distinct melt episodes with larger discharges were recorded on 13 June (0.81 m$^3$ s$^{-1}$), 19 June (0.94 m$^3$ s$^{-1}$), and 16 July (0.93 m$^3$ s$^{-1}$). These peak flows occurred at around 15:00 local time and were almost double the melt season average of daily maximum discharge (0.5 m$^3$ s$^{-1}$). The timing of these melt episodes corresponds with periods of anomalously high river discharge observed ∼ 35 km downstream in the Watson River, Kangerlussuaq (van As et al., 2018).

Daily maximum discharge varies from 0.05–0.94 m$^3$ s$^{-1}$ through the season, with the highest values around the three melt episodes (Fig. 5b). Daily minimum discharge has much less variability over the melt season than daily maximum discharge but exhibits two occurrences with anomalously larger flow on 23 June and 19 July (Fig. 5b). During melt episodes, these positive anomalies in daily minimum discharge follow a steep decrease in daily maximum discharge, meaning the anomalously large low flows at night follow a dip in daytime streamflow. Between the second and third melt episodes, nighttime low-flow discharge occasionally falls as low as 0.002 m$^3$ s$^{-1}$ but remains above 0.04 m$^3$ s$^{-1}$ after the third episode. Finally, the diurnal amplitude (daily maximum minus daily minimum discharge) tracks daily maximum discharge except for in the second and third melt episodes due to large daily minimum discharge at those times (Fig. 5c). After the third melt episode, there is a steady decline in diurnal amplitude from 0.64 m$^3$ s$^{-1}$ on 21 July to 0.33 m$^3$ s$^{-1}$ on 13 August.

The daily mean discharge varies from 0.02–0.51 m$^3$ s$^{-1}$ over the 62 d (Fig. 6a) and co-varies over the melt season with daily maximum discharge except during the second and third melt episodes (Fig. 5b). The daily mean discharge peaks on 19 July, 3 d after the second-largest melt episode in hourly discharge. In contrast, the second-largest peak in daily mean flow occurs on 20 June, which is on the same day as the largest episode in hourly discharge. In both cases, hourly maximum discharge is accompanied by several days of very high daily minimum flows (Fig. 5b), which explains the discrepancy between the timing of the daily mean and daily maximum episodes.

### 4.2 Surface energy balance

Throughout the season, net shortwave radiation exceeds all other surface energy fluxes and thus is the primary driver of stream discharge (Fig. 6b). However, the second and third melt episodes coincide with peak longwave radiation (65 W m$^{-2}$ increase compared to before the episodes) and turbulent heat fluxes (40–80 W m$^{-2}$ increase) along with a drop in shortwave radiation (110–120 W m$^{-2}$ decrease) (Fig. 6). Thus, during high-melt episodes, longwave radiation and turbulent heat fluxes become more pronounced drivers of streamflow. Among all energy fluxes, sensible heat flux correlates the most with daily mean discharge ($R = 0.88$; $p$ value $< 0.01$). During the third melt episode, the hourly peak discharge coincides with a peak in shortwave radiation on 16 July (Fig. 7a). However, the peak daily mean discharge occurs 3 d later on 19 July 2016 due to high net longwave radiation and turbulent heat fluxes from 16–20 July (Fig. 6). This episode of high net longwave radiation was caused by overcast conditions (with cloud cover consistently greater than 0.4, except for a couple of hours throughout a 96 h period; Fig. 7b) and resulted in large low flow at night (Fig. 7a). This consistently large low flow persisted from 17 to 20 July (Fig. 7a) and resulted in a peak of average daily discharge on 19 July (0.51 m$^3$ s$^{-1}$) (Fig. 6a). This can also be seen in the hourly variation in surface energy balance components and stream discharge (Fig. 7a). Between 17–20 July, nighttime streamflow is much higher than before and after the third melt episode (Fig. 7a) and coincides with increased net longwave radiation. While a dip in shortwave radiation on 18 July decreases the high flow during the day, the low flow during the night increases due to a spike in net longwave radiation (Fig. 7a).

To further examine each energy balance components' contribution to stream discharge, we aggregated components for the second and third melt episodes and compared them to data spanning the entire melt season (Fig. 8). Contribution of individual components is estimated as a ratio of the total melt energy and is described as the proportion of melt energy. The shortwave radiation proportion of melt energy fell by 40 % from a melt proportion of 1.13 to 0.73 during the melt episodes. Simultaneously, the contribution of longwave radiation and turbulent heat fluxes increased during those days. The longwave radiation's proportion of melt energy increased from a melt season average of −0.32 to −0.08 during

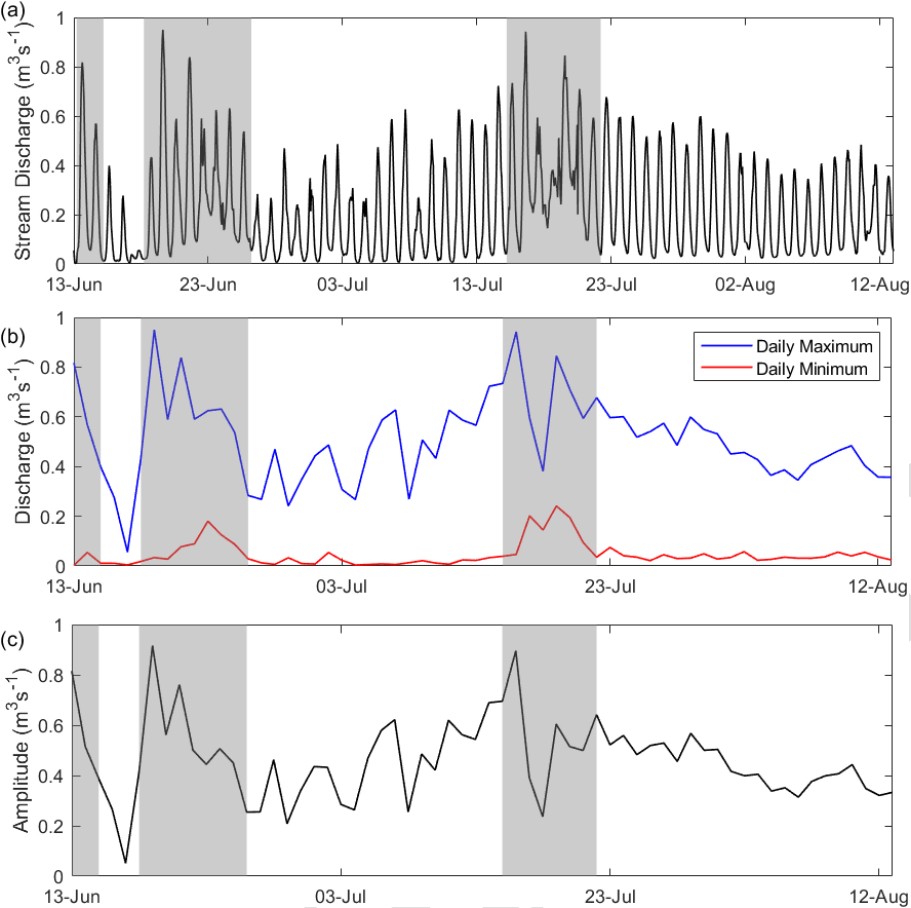

**Figure 5. (a)** Hourly stream discharge generated using the rating curve (Fig. 4) over 62 d of the melt season. **(b)** Daily maximum (in blue) and daily minimum (in red) discharge, calculated from the hourly discharge. **(c)** Amplitude (daily maximum minus daily minimum) of the stream discharge. Three large melt episodes are shown in grey-shaded regions.

the peak flow days, corresponding to an increase in contribution by 24 % (Fig. 8). The sensible and latent heat fluxes' proportion of melt energy increased from 0.28 to 0.34 (6 %) and from −0.4 to 0.6 (10 %) respectively during the melt episodes (Fig. 8).

### 4.3 Timing of daily maximum discharge

To examine if the transport of meltwater from its production on the ice sheet surface to the discharge observation site varies over the season, we calculated the time to daily maximum discharge, following the "time-to-peak" methodology in traditional terrestrial hydrology (Chow, 1964). As the season progresses, the timing of daily maximum discharge will reflect temporary changes in melt storage within weathering crust and meltwater transport efficiency. In contrast, during clear-sky days, when solar radiation drives melt, the timing of daily maximum surface meltwater production is not expected to change over the season and is proportional to solar noon. Therefore, during the clear-sky days, variability in the timing of daily maximum discharge can be attributed to net-

work storage and transport efficiency as opposed to during non-clear-sky days with noise in the signal due to the variation in incoming solar radiation and clouds (Fig. S2). For example, when cloud cover greater than 0.6 persists for longer than 3–4 h during the middle of the day (10:00–16:00 local time), peak discharge occurs earlier in the day, around noon to 13:00 local time. However, if cloud cover persists for less than 3 h around midday, peak discharge occurs later in the day between 15:00–17:00 local time.

While the incoming solar radiation peaks at solar noon (around 13:15 local time), the timing of the daily maximum discharge varies from 16:00 in late June to 14:00 in late July (Fig. 9a). In other words, the peak time lag between the solar and discharge peaks changes from 3 to 1 h from 30 June to 31 July and has a statistically significant negative trend ($R^2 = 0.79$; $p$ value $<0.01$). After 31 July, the peak time lag abruptly shifts back to early melt season conditions of a 3 h time lag. This shift in the peak time lag coincides with the sudden decrease in daily mean temperatures from 4.3 °C on 31 July to 2.5 °C on 3 August, with daily minimum temperatures dropping down to 1 °C (Fig. 9b). These air temperature

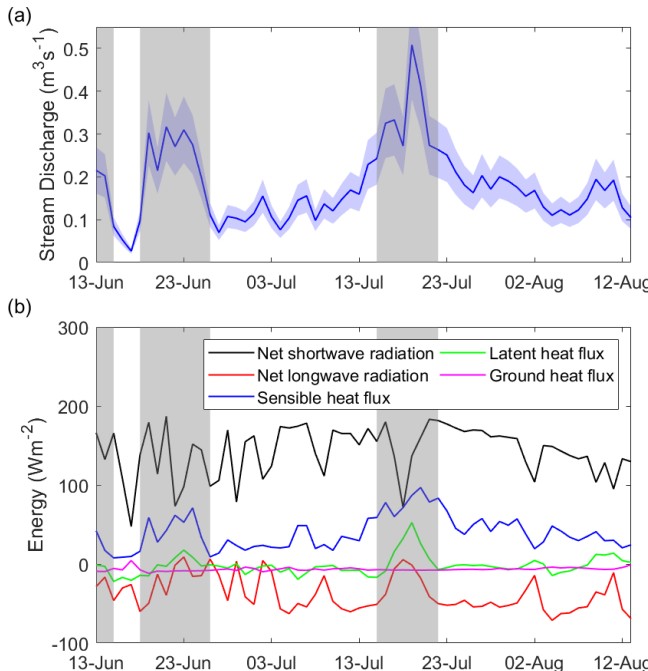

**Figure 6. (a)** Daily discharge calculated by averaging the hourly discharge from Fig. 5a. The uncertainty in the daily mean discharge generated by averaging hourly discharge, $X_{dm}$, is shown in the blue-shaded area (see Appendix A for uncertainty calculations). **(b)** Surface energy components – net shortwave radiation (black), net longwave radiation (red), sensible heat flux (blue), latent heat flux (green), and ground heat flux (magenta). Large melt episodes are shown in grey-shaded regions.

measurements were collected at 2 m above the ice surface, and therefore the skin temperatures are expected to be below freezing, causing meltwater delivery to the channels to slow down. With below-freezing temperatures, there is likely an increase in Manning's $n$ coefficient (i.e., quantifying channel roughness and friction; Chow, 1964) as frozen ice features pose an impedance to flow, in turn lowering the streams' conveyance. In addition to the change in temperature, a sudden drop in water level in the cryoconite hole coincides with the drop in temperature and abrupt shift in the time of daily maximum discharge (Fig. 9c).

### 4.4 Hydraulic geometry

Hydraulic geometry parameters are determined by generating a power law between stream discharge and width ($R^2 = 0.87$; $p$ value $< 0.01$), depth ($R^2 = 0.94$; $p$ value $< 0.01$), and velocity ($R^2 = 0.88$; $p$ value $< 0.01$) (Eqs. 2–4 respectively). For the 660 catchment, the exponents $b$, $f$, and $m$ are 0.19, 0.39, and 0.37 respectively, and coefficients $a$, $c$, and $k$ are 3.44, 0.54, and 0.63 respectively. In theory the sum of the exponents of these power laws, representing the sensitivity of discharge to the individual variable, equals 1, and the product of the coefficients must also equal 1 (Leopold and

Maddock, 1953). But, in practice, due to measurement error and when $R^2 < 1.0$, i.e., the power law does not perfectly describe the data, we can expect deviations from 1. In our study the sum of exponents equals 0.95 and the product of coefficients equals 1.17.

Variation in hydraulic geometry parameters was investigated over three time periods of the melt season, 17 June–20 July, 26–27 July, and 4 August. Though these time periods have different sample sizes ($N$ is 15, 27, and 4 respectively), the $R^2$ values for all the parameters are greater than 0.89 ($p$ value $< 0.01$) for both before and after melt episodes. However, the parameters in the August sample are not significant with $R^2$ values 0.51, 0.52, and 0.01 ($p$ value equal to 0.25, 0.32, and 0.96) for width, depth, and velocity exponents respectively due to a small sample size. Analysis over different time periods of the melt season shows a dramatic drop in the velocity exponent ($m$) on 4 August compared to earlier in the season (Fig. 10). Velocity had a higher exponent (meaning stronger relation to $Q$) compared to other parameters until early August; then there is a shift to depth having a stronger relation to $Q$, thereby reducing the dependency of $Q$ on velocity. While the width exponent ($b$) ranges between 0.1–0.2 throughout the season, the depth exponent increases gradually from 0.3 before the July melt episode to 0.4 after the melt episode to 0.5 in early August.

## 5 Discussion

Here, we present a 62 d time series of supraglacial stream discharge (13 June–13 August 2016). We find strong diurnal variability in stream discharge, similarly to previous in situ studies of supraglacial streamflow (Holmes, 1955; Knighton, 1981; Marston, 1983; Mernild et al., 2006; McGrath et al., 2011; Chandler et al., 2013, 2021; Wadham et al., 2016; Smith et al., 2017, 2021) despite different locations. At our study site, the 660 catchment (TS5 $0.6 \pm 0.03 \, \text{km}^2$), diurnal variability ranges from close to zero to as much as $0.95 \, \text{m}^3 \, \text{s}^{-1}$ with daily maximum discharge occurring between 14:00–16:00 local time throughout the study period. Both diurnal variability and the time of maximum discharge are comparable to McGrath et al. (2011), who documented diurnal variability of $0.017–0.54 \, \text{m}^3 \, \text{s}^{-1}$ with a daily maximum discharge at 16:45 local time over a catchment (in the Sermeq Avannarleq ablation zone in central-west Greenland) of area $1.14 \pm 0.06 \, \text{km}^2$ from 3–17 August 2009 (Table 1). Marston (1983) also finds a similar range of discharge varying from close to zero to $0.23 \, \text{m}^3 \, \text{s}^{-1}$ with a daily maximum discharge occurring between 14:00–16:00 local time on the Juneau Icefield around late July. Mernild et al. (2006) and Chandler et al. (2013) report diurnal variability up to 10 times larger than at the 660 catchment and a daily maximum discharge occurring between 14:00–18:00 local time from two catchments larger than the 660 catchment. The oldest study we are aware of, Holmes (1955), reports supraglacial stream

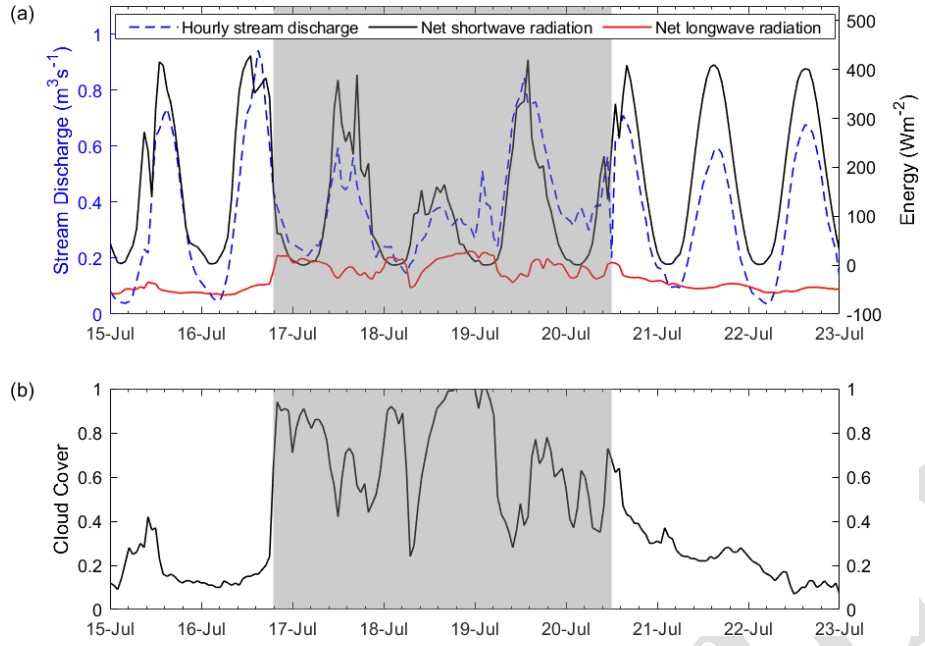

**Figure 7. (a)** Diurnal fluctuations in stream discharge on the left $y$ axis (dashed blue line) and surface energy balance components on the right $y$ axis, net shortwave radiation in black and net longwave radiation in red from 15–22 July. **(b)** Cloud cover at the KAN_L station from 15–22 July. The large daily minimum period (a subset of the third melt episode) with cloud cover consistently greater than 0.4, except for a couple of hours throughout a 96 h period, is shown in the grey-shaded region.

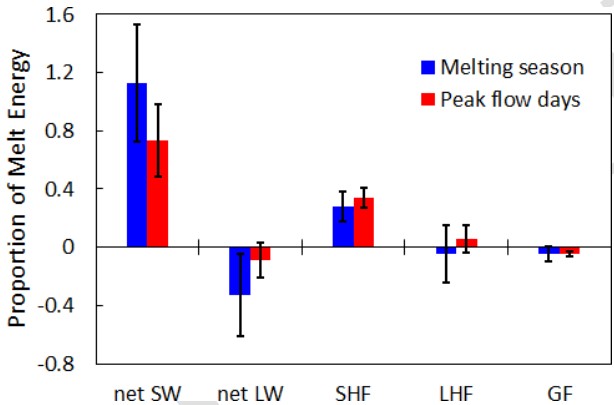

**Figure 8.** Proportion of melt energy (ratio of each component to total melt energy) for the whole melt season in blue and for the days during the melt episodes only in red. Here, peak flow days include days from the second melt episode (19–23 June) and the third melt episode (16–20 July). Error bars represent standard deviation of each sample. net SW – net shortwave radiation; net LW – net longwave radiation; SHF – sensible heat flux; LHF – latent heat flux; GF – ground heat flux.

discharge of 0.14–TS6 5 m$^3$ s$^{-1}$ (about 5 times larger than at the 660 catchment) at a catchment of 25–50 km$^2$ (40–80 times larger than the 660 catchment) with a daily maximum discharge occurring between 16:00–20:00 local time in southwest Greenland. In an even larger catchment (60–63 km$^2$, $\sim$ 100 times larger than the 660 catchment), Smith

et al. (2017, 2021) documented that the daily maximum discharge occurred between 18:00–20:00 and 20:30–22:40 local time, and discharge varied between 4.6–26.7 and 5.8–37.6 m$^3$ s$^{-1}$ (Table 1) in late July 2015 and early July 2016 respectively. Synthesizing all studies providing time series of stream discharge (Marston, 1983; Mernild et al., 2006; McGrath et al., 2011; Chandler et al., 2013; Smith et al., 2017, 2021), the lag between solar noon and daily maximum discharge, i.e., peak time lag, is larger for the larger magnitude of stream discharge. This can be explained by the fact that when runoff is generated over a larger catchment area, the distance of surface routing increases and thus delays daily maximum discharge at the catchment outlet (Yang and Smith, 2016; Smith et al., 2017; King, 2018).

Over the 62 d study period, while net shortwave radiation provides the majority of melt energy and is the primary driver of streamflow, net longwave radiation and turbulent heat fluxes (sensible and latent) become more dominant melt drivers during the three melt episodes (Fig. 8). These findings disagree with studies suggesting that overcast conditions, resulting in lower incoming solar radiation, reduce surface melt in the ablation zone (Hofer et al., 2017; Izeboud et al., 2020) but agree with Greenland-wide studies identifying a link between longwave radiation and enhanced surface melting (Van Tricht et al., 2016; Gallagher et al., 2020). Furthermore, out of all energy balance components (e.g., net shortwave radiation; $R^2 = 0.23$; $p$ value $= 0.035$), the daily average sensible heat flux has the highest correlation with stream discharge

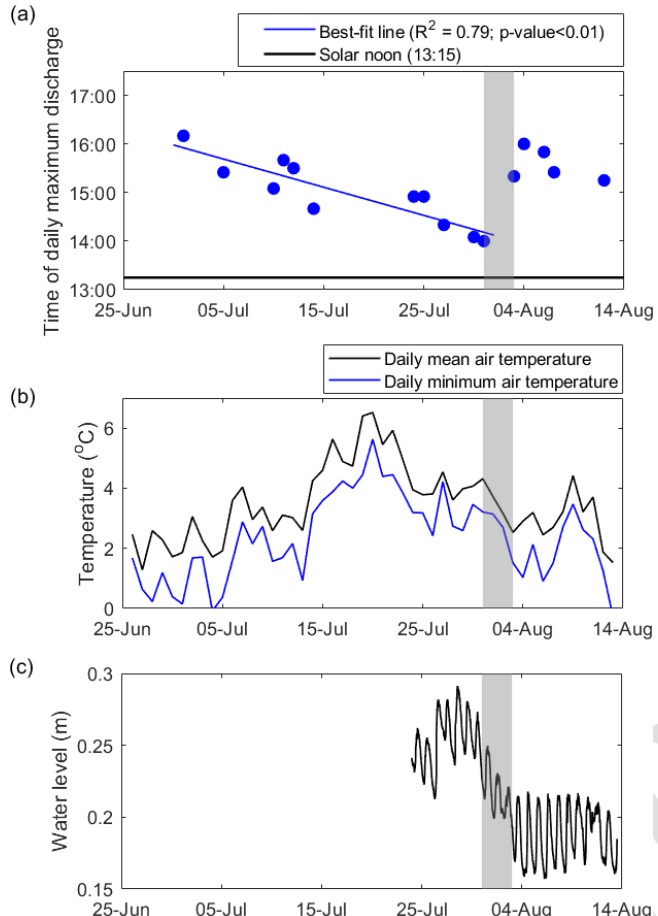

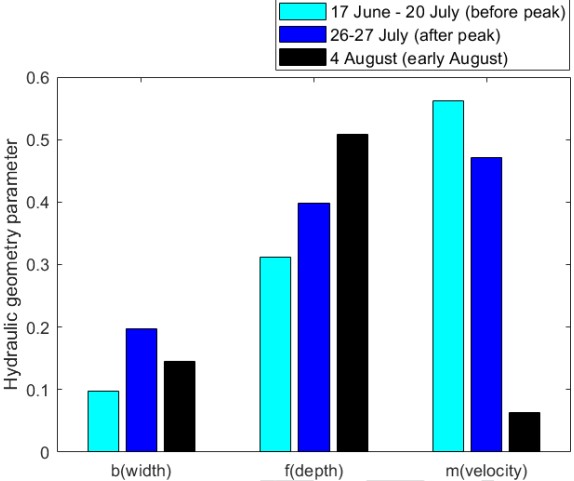

**Figure 10.** Hydraulic geometry parameters calculated for three different times, (1) bulk of melt season before peak (melt episode in July) – 17 June–20 July, (2) after the peak (melt episode in July) but before an increase in peak timing – 26–27 July, and (3) after the peak (melt episode in July) and after an increase in peak timing (in early August) – 4 August.

**Figure 9. (a)** Time to daily maximum discharge for clear-sky days. Clear-sky days were identified as days with incoming solar radiation (from KAN_L AWS data) with a smooth diurnal cycle and lacking short-term, hourly fluctuations from varying cloud cover. **(b)** Daily mean air temperature from KAN_L AWS. The period where change in the time of daily maximum discharge coincides with a sudden drop in temperature from 1–3 August is shown in the grey-shaded region. **(c)** Water level in a cryoconite hole. Measurements were made with the same type of Solinst Levelogger® as used to measure the channel water level.

($R^2 = 0.83$; $p$ value $< 0.01$). This contribution of sensible heat flux is consistent with Fausto et al. (2016), who show that peak melting occurs at times with anomalously large turbulent energy fluxes. Correlating energy components with stream discharge without a time lag is justified here due to the quick routing in this catchment (1–3 h). Though previous studies have shown a similar link between sensible heat flux and episodes of intense melting (van As et al., 2012; Fausto et al., 2016; Wang et al., 2021), all of them rely on model-simulated melt using local weather station data, while we are using stream discharge in this study. Net shortwave radiation has the largest contribution to total meltwater production, which is reduced by 40 % (melt proportion decreases from 1.13 to 0.73) during the melt episodes. This reduction in contribution to melt is compensated for by net longwave radiation and sensible and latent heat fluxes, which increased by 24 %, 6 %, and 10 % (corresponding to melt proportion increase from −0.032 to −0.08, 0.28 to 0.34, and −0.4 to 0.6) respectively during the melt episodes.

For the 660 catchment, the peak time lag, i.e., the lag between peak incoming solar radiation and daily maximum discharge, decreases through the melt season from 3 h in late June to 1 h in late July. The shorter peak time lag compared to previous studies (3–9.5 h; Holmes, 1955; Mernild et al., 2006; McGrath et al., 2011; Chandler et al., 2013; Smith et al., 2017) is likely due to the smaller size of the 660 catchment. The reduced peak time lag through the melt season is consistent with the change observed by Mernild et al. (2006) from 5–7 h in May to 3–4 h in August, attributed to changes in weathering crust structure. Furthermore, at the 660 catchment, in early August, the peak time lag abruptly increases to the initial melt season conditions and stabilizes at 3 h, coinciding with a sudden drop in air temperature from 4.3 to 2.5 °C from 31 July to 3 August (Fig. 9b). The time of daily maximum discharge is driven by the catchment's ability to evacuate water, which in turn depends on the rate of meltwater transport in the stream channels and the proportion of the transport distance that is dictated by porous media flow (i.e., non-channelized flow through weathering crust and over bare ice). Meltwater trapped in weathering crust also can decrease drainage efficiency. This drainage efficiency depends on the geometry of the channel network, the hydraulic conductivity and storage capacity of weathering crust, and the frictional coefficient of the streambed (Karlstrom, 2014; Gleason et al., 2015; Cooper et al., 2018). We hypothesize

**Table 1.** Table of measured supraglacial streamflow, width, and catchment size from this and previous studies. "Width" denotes stream width, and "Lag" is the time of peak discharge after solar noon. NA – not available.

| Source | Location | Time | Discharge $(m^3\,s^{-1})$ | Width (m) | Catchment area $(km^2)$ | Lag (hours) |
|---|---|---|---|---|---|---|
| Holmes et al. (1955) | Alpha River, Project Mint Julep, southwest Greenland | 21 July–15 August 1953 | $\sim 0.14$–5.11 | 9 | 25–50 | 3–7 |
| Knighton (1981) | Austre Okstindbreen, Norway | NA | 0.005–0.02 | 0.2–0.5 | NA | NA |
| Marston (1983) | Juneau Icefield | 28 July–2 August 1983 | $\sim 0$–0.24** | 0.7–1.6 | NA | 2–3 |
| Mernild et al. (2006) | Mittivakkat Glacier, southeast Greenland | 16–19 August 2004, 15–18 June 2005 | 5–10 | NA | 18.4 | 3–4 (August), 5–7 (May) |
| McGrath et al. (2011) | West Greenland | 3–17 August 2009 | 0.017–0.54 | 0–2.5 | $1.14 \pm 0.06$ | 3–4 |
| Chandler et al. (2013) | Moulin L41 – internally drained catchment (part of Leverett catchment), southwest Greenland | 29 June–7 July & 17–20 August 2011 | $\sim 0.1$–8** | NA | NA | 3–6 (July) 6–8 (August) |
| Wadham et al. (2016) | Leverett Glacier, southwest Greenland | 16 June–10 August | $\sim 0$–12** | NA | NA | NA |
| Gleason et al. (2016) & Smith et al. (2015)* | Southwest Greenland | July–August 2012 | 0.006–0.402 (small streams), 4.58–23.12 (large streams) | 0.2–3.84 (small streams), 7.19–20.62 (large streams) | NA | NA |
| Smith et al. (2017) | Southwest Greenland | 20–23 July 2015 | 4.61–26.73 | 6–19 | 63.1 | 4–6 |
| Smith et al. (2021) | Southwest Greenland | 6–13 July 2016 | 5.75–37.61 | up to $\sim 30$ (est.) | 60.2 | 6.5–7.5 |
| 660 Catchment (this study) | Southwest Greenland | 13 June–13 August 2016 | 0.002–0.95 | 1.6–3.2 | 0.6 | 1–3 |

* Smith et al. (2015) and Gleason et al. (2016) have common data sets. The range of width and discharge are the same for both the studies. However, Gleason et al. (2016) primarily discussed the hydraulics of these streams. Therefore, this data set is mentioned as that of Gleason et al. (2016) in the discussion. ** Discharge from these studies is visually estimated from their figures and therefore approximated to the closest first decimal place. The lower bound of stream discharge in Marston (1983) is taken as zero as the value was very small and close to zero in the figure presented.

that, as the melt season progresses to peak discharge in July and meltwater production increases, the catchment increases the proportion of channelized flow compared to porous media flow and stream density. Yang et al. (2018) demonstrated using hydrologic modeling that increased stream density and the importance of channelized flow result in an increase in the drainage efficiency, larger discharge amplitude, and earlier time of peak discharge. The high-melt episodes with increased longwave radiation also contribute to the drainage efficiency by decreasing weathering crust storage capacity (Takeuchi, 2000). However, in early August, a steep drop in the water level inside the weathering crust layer coincides with a drop in air temperature. With the nighttime air temperatures close to $0\,°C$, the streambed likely partially freezes. This ice formation impedes flow, likely causing an increase in the Manning's $n$ coefficient of the stream channel. This in turn causes discharge to be more regulated by changes in cross-sectional area rather than in velocity, which is seen

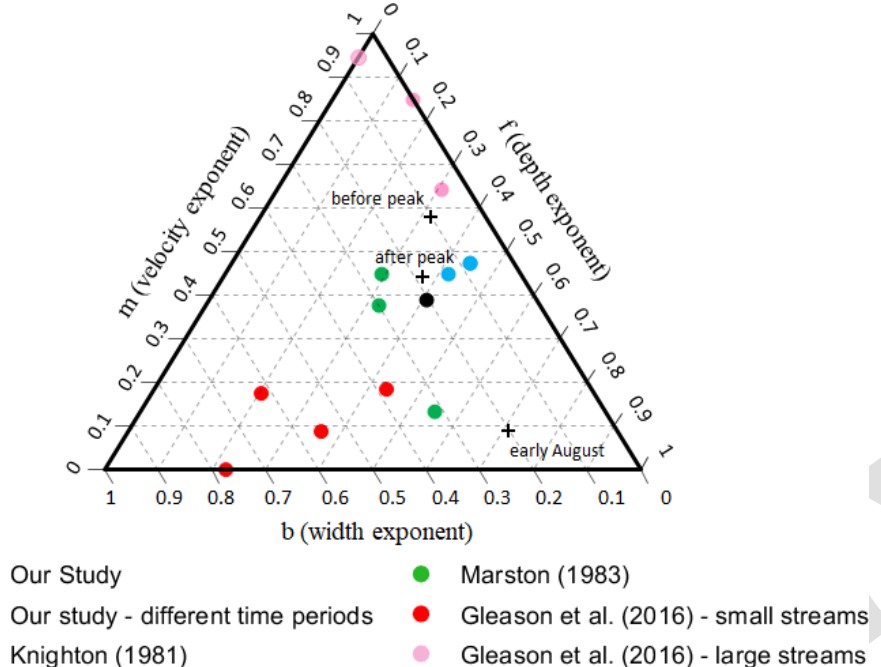

**Figure 11.** Ternary diagram comparing $m$, $f$, and $b$ parameters from this study (in black; whole season shown by solid dot; different time periods shown by "+" with before peak being 17 June–20 July, after peak being 26–27 July, and early August being 4 August) with previous work by Knighton (1981) in blue, Marston (1983) in green, and Gleason et al. (2016) large streams in pink and small streams in red. Where $m + f + b$ exceeded unity, parameters were adjusted to unity.

as a sharp drop in meters (velocity exponent) in early August (Fig. 10). This also likely coincides with the stream network switching back to a higher proportion of porous media flows, which have CES longer transport distances and there-
5 fore will increase the peak time lag. Additionally, similarly to the streambed, the weathering crust layer is likely at freezing temperature in August, which thus results in increased interstitial freezing and a decrease in hydraulic conductivity (Cooper, 2020).
Our work confirms the findings by Gleason et al. (2016) that hydraulic geometry parameters cannot be generalized for supraglacial rivers in Greenland, despite having a common ice substrate. This study furthers Gleason's conclusions by analyzing streams closer to the ice edge, showing that hy-
draulic parameters are still highly spatially and temporally variable across the ice sheet and may vary over a melt season. Comparing our data with parameters from previous studies (Knighton, 1981; Marston, 1983; Gleason et al., 2016) in a ternary diagram reveals three clusters (Fig. 11). These three
clusters can be grouped based on their $b$ values (width exponent). The first cluster has high $b$ values ($b \geq 0.35$) and includes the downstream stations of Gleason et al. (2016), which are in smaller streams with discharge varying between 0.006 and 0.402 m$^3$ s$^{-1}$. The second cluster has low
$b$ values ($b \leq 0.05$) and includes at-a-station data from Gleason et al. (2016), which are from larger streams with discharge varying between 4.58 and 23.12 m$^3$ s$^{-1}$. Finally, the

third cluster has moderate $b$ values ($0.05 < b < 0.35$) and includes this study and Knighton (1981) and Marston (1983). Though the discharge from Knighton (1981) is 2 orders of
30 magnitude smaller than ours, Marston (1983) has similar discharge values (Table 1). The streams with discharge of the same order of magnitude i.e., varying between 0–1 m$^3$ s$^{-1}$, from Knighton (1981), Marston (1983), and the current study show moderate sensitivity to stream width (moderate values
of $b$), and streams with higher magnitude of discharge show very small sensitivity to stream width (smaller values of $b$). However, small streams from Gleason et al. (2016) do not concur with this generalization and show a high sensitivity to stream width (large $b$ values). Changing hydraulic geom-
etry parameters over the melt season may explain some of the differences between the studies as our parameters determined during the main melt season (17 June to 20 July) are close to large streams parameters from Gleason et al. (2016) which were collected in middle–late July. On the other hand,
our parameters determined at the end of the melt season (4 August) approach small stream parameters from Gleason et al. (2016) which were collected around mid-August. These parameters seem to be more dependent on the time of the melt season than the location or size of the streams.
Our analysis of a melt season-long record of streamflow and its drivers has several implications for large-scale Greenland ice sheet hydrology. First, we find that longwave radiation and turbulent fluxes have an increased contribution by

24 % and 16 % respectively in governing stream discharge during the melt episodes. Given that several regional climate models underestimate turbulent fluxes, they will also underestimate melt episodes (van den Broeke et al., 2011; Fausto et al., 2016). Since one of the most widely used methods to estimate surface runoff from the entire Greenland ice sheet is through regional climate/surface mass balance models (Cullather et al., 2016; Fettweis et al., 2017; Mernild et al., 2018; Noël et al., 2018), underestimating turbulent heat fluxes also underestimates runoff. Second, with the lack of long-term records of supraglacial stream discharge over the Greenland ice sheet, the 62 d long time series could expand climate/surface mass balance models validation capability. For context, a recent study validating model-simulated runoff using field observations of supraglacial streamflow covers just 3 d (Smith et al., 2017). Lastly, understanding the evolution of the accurate timing and magnitude of peak discharge throughout the melt season may aid future studies about the influence of peak discharge on CE9 subglacial water pressure and ice velocities.

## 6 Conclusions

We present one of the longest records of Greenland supraglacial stream discharge, spanning 62 d of the 2016 melt season for a TS7 0.6 km$^2$ catchment in southwest Greenland. These observations could be used in validating regional climate models, currently the best tools to estimate surface runoff from the entire Greenland ice sheet. The observed stream discharges vary both diurnally and over the melt season. Our record includes three distinct episodes of large discharge, one from 13–14 June, a second from 19–25 June, and a third from 15–21 July. The daily maximum discharge and amplitude show similar diurnal and melt season variations except during the second and third melt episodes, when large nighttime melting reduced the daily amplitude. During the third melt episode, the large nighttime flows (i.e., daily minimums) drive the peak in average daily discharge on 19 July (both the day- and nighttime have continuous high flow between 16–19 July). The stream discharge is primarily driven by net shortwave radiation through the melt season, except during the high-melt episodes when net longwave radiation and turbulent heat fluxes show an increased contribution (24 % and 16 % respectively) to melt energy. The peak time lag, i.e., the lag between the time of daily maximum discharge and solar noon (close to the time of daily maximum melt on clear-sky days), varies through the melt season from 3 h in late June to 1 h in late July and goes back to 3 h in early August. The abrupt shift in the peak time lag in early August is attributed to a sudden drop in air temperature, steep decrease in temporary water storage in weathering crust, and a change in hydraulic geometry. On the other hand, the gradual decrease in peak time lag through the melt season could be due to the expansion of the stream network and increased ra-

tio of channelized to porous flow. Further work is required to reveal if the rapid shift in the timing of peak discharge, which we observe at the 660 catchment, also takes place across Greenland supraglacial streams, over larger catchments and at higher elevations compared to our study, and to analyze the influence of stream network development on the timing and magnitude of peak discharge.

## Appendix A: Streamflow uncertainty

The uncertainty in the 46 discrete discharge observations due to seven different types of measurement error in velocity and stage height was calculated following Herschy (2002) and WMO (2010) and assumes that segment discharges and standard uncertainties are approximately equal in each of the segments in the cross-section:

$$U_{me}(Q) =$$

$$K\sqrt{u_m^2 + u_s^2 + \frac{1}{M}\left\{u_b^2 + u_d^2 + u_p^2 + u_c^2 + u_e^2\right\}}, \quad (A1)$$

where $U_{me}$ is uncertainty due to measurement errors, $Q$ is discharge, $K$ is the so-called coverage factor ($K = 2$ gives the 95 % confidence interval), $u_m$ is uncertainty in the determination of mean velocity for the number of verticals ($M$), $u_s$ is uncertainty due to calibration errors, $u_b$ is uncertainty in width, $u_d$ is uncertainty in depth, $u_p$ is uncertainty in the determination of mean velocity for the number of points in the vertical, $u_c$ is uncertainty in the determination of mean velocity for the current meter rating, and $u_e$ is uncertainty in the determination of mean velocity for the time of exposure. A summary of all nomenclature used in this study is found in Table A1.

Using Eq. (A1) and literature values for the seven measurement errors (Table A2), we find $U_{me} = 10.8$ %.

The uncertainty in hourly stream discharge due to the rating curve was determined with a statistical method by calculating the standard error of estimate, $S_e$, where the quadratic rating curve was linearized with a logarithmic transformation following Herschy (1994):

$$S_e = \pm t\sqrt{\frac{\sum_{i=1}^n (\ln Q_i - \ln Q_c)^2}{n - 2}}, \quad (A2)$$

where $t$ is Student's $t$ correction (for 95 % confidence $t = 2$), $n$ is the number of discharge measurements, $Q_i$ is discharge measured, and $Q_c$ is discharge estimated using a rating curve.

Using Eq. (A2), the uncertainty in hourly stream discharge due to the rating curve, $U_{RC}$, is calculated at 17 %.

Lastly, the uncertainty in daily mean discharge due to the averaging of hourly data is estimated in two steps using the methodology of Dymond and Christain (1982). First, $S_{mr}$, the standard error of the daily mean, is calculated using a

**Table A1.** Nomenclature used in this paper.

| Symbol | Unit | Description |
|--------|------|-------------|
| $H$ | m | Water level/stage height |
| $Q$ | $m^3 s^{-1}$ | Discharge |
| $Q_i$ | $m^3 s^{-1}$ | Measured stream discharge |
| $Q_c$ | $m^3 s^{-1}$ | Estimated stream discharge from a rating curve |
| $w$ | m | Width |
| $d$ | m | Depth |
| $v$ | $m s^{-1}$ | Velocity |
| $\alpha$ | m | Datum correction (stage at zero flow) |
| $\beta$ | – | Constant (exponent in the rating curve equation) |
| $q$ | – | Constant (coefficient multiplying stage in the rating curve equation) |
| $a$ | – | Width hydraulic geometry coefficient |
| $c$ | – | Depth hydraulic geometry coefficient |
| $k$ | – | Velocity hydraulic geometry coefficient |
| $b$ | – | Width hydraulic geometry exponent |
| $f$ | – | Depth hydraulic geometry exponent |
| $m$ | – | Velocity hydraulic geometry exponent |
| $N$ | – | Number of data points in a sample |
| $n$ | – | Number of discharge measurements |
| $M$ | – | Number of verticals |
| $K$ | – | Coverage factor ($k = 2$ for 95 % CI) |
| $t$ | – | Student's $t$ correction (for 95 % confidence $t = 2$) |
| $u_m$ | % | Uncertainty in determination of mean velocity for number of verticals |
| $u_s$ | % | Uncertainty due to calibration errors |
| $u_b$ | % | Uncertainty in width |
| $u_d$ | % | Uncertainty in depth |
| $u_p$ | % | Uncertainty in determination of mean velocity for number of points in the vertical |
| $u_c$ | % | Uncertainty in determination of mean velocity for current meter rating |
| $u_e$ | % | Uncertainty in determination of mean velocity for time of exposure |
| $U_{RC}$ | % | Uncertainty in hourly stream discharge due to the rating curve |
| $U_{me}$ | % | Uncertainty due to measurement errors |
| $U_{in}$ | % | Uncertainty due to streambed incision |
| $X_{dm}$ | % | Uncertainty in the daily mean discharge |
| $S_e$ | % | Standard error of estimate |
| $S_{mr}$ | % | Standard error of mean |
| $S_{wl}$ | % | Standard error of log of water level measurement |

logarithmic transformation of discharge, $Q$ from Eq. (1), in order to make it a linear relation:

$$S_{mr} =$$

$$\pm t S_e \left( \frac{1}{N} + \frac{(\ln\ln(h+\alpha) - \underline{\ln(h+\alpha)})^2}{\sum (\ln\ln(h+\alpha) - \underline{\ln(h+\alpha)})^2} \right)^{1/2}, \quad \text{(A3)}$$

**Table A2.** Standard uncertainties in a single measurement of stream discharge due to seven measurement errors, using empirical values from WMO (2010). The final uncertainty due to measurement error is calculated with a 95 % CI using the empirical values from this table. Please refer to Table A1 for nomenclature.

| Uncertainties | Values used in this study |
|---------------|---------------------------|
| $u_m$ | 4.5 % (for a minimum of 10 verticals) |
| $u_s$ | 1 % |
| $u_b$ | 0.15 % (for width range 0–100 m) |
| $u_d$ | 0.65 % (for depth range 0.4–6 m) |
| $u_p$ | 7.5 % (number of verticals is 1) |
| $u_c$ | 1 % (for average velocity of around $0.25\,m\,s^{-1}$) |
| $u_e$ | 4 % (for time of exposure between 30–60 s) |
| $M$ | 10 (number of verticals varied between 10–16) |
| $K$ | 2 (for 95 % CI) |

where $N$ is the number of data points in the sample (here, 24 samples in a day). Second, $X_{dm}$ is uncertainty in the daily mean discharge and is calculated as

$$X_{dm} = \frac{1}{N} \sum_{i=1}^{N} \sqrt{S_{mr}^2 + \beta^2 S_{wl}^2} Q_i , \quad \text{(A4)}$$

where $S_{wl}$ is the standard error of the log of water level measurement (calculated using Eq. A2). Using Eqs. (A3) and (A4), the uncertainty in the daily mean discharge, $X_{dm}$, is calculated at 25 %. All estimated uncertainties ($U_{me}$, $U_{RC}$, and $X_{dm}$) are expressed as the 95 % confidence level.

*Data availability.* KAN_L weather station data from the Programme for Monitoring of the Greenland Ice Sheet (PROMICE) and the Greenland Analogue Project (GAP) were provided by the Geological Survey of Denmark and Greenland (GEUS) at http://www.promice.dk (GEUS, 2022 TS8). CE10 All other data are available at the Arctic Data Center (https://doi.org/10.18739/A2XW47X5F TS9, Muthyala et al., 2022).

*Supplement.* The supplement related to this article is available online at: https://doi.org/10.5194/tc-16-1-2022-supplement.

*Author contributions.* RM performed data analysis and wrote the manuscript with support from AKR. RM and AKR conceived and planned the study. SZL and MGC carried out the majority of field data collection, with support from RM, AKR, and SWC. DvA performed surface energy balance modeling. All authors discussed the results and contributed to the final paper.

*Competing interests.* The contact author has declared that neither they nor their co-authors have any competing interests.

*Acknowledgements.* Funding for this work comes from the NASA Cryospheric Sciences Program (award nos. 80NSSC19K0942 and NNX14AH93G) managed by Thorsten Markus. Sasha Z. Leidman was funded by the NSF Graduate Research Fellowships Program. QGIS was used to prepare maps. The Polar Geospatial Center's Arctic DEMs were used in this study. Geospatial support for this work provided by the Polar Geospatial Center under NSF-OPP awards 1043681 and 1559691. We thank Kyle Mattingly, Jing Xiao, Isatis Cintron, David Chandler, and the three anonymous reviewers for constructive feedback on the manuscript.

*Financial support.* This research has been supported by the Earth Science Division (grant nos. 80NSSC19K0942 and NNX14AH93G).

*Review statement.* This paper was edited by Brice Noël and reviewed by three anonymous referees.

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

## Remarks from the language copy-editor

## Remarks from the typesetter