# Peer review of "Supraglacial Streamflow and Meteorological Drivers from Southwest Greenland"

_The Cryosphere, 2020_

## Referee Comment (RC1) · Anonymous Referee #1 · 22 Dec 2020

This manuscript describes the 46 discharge measurements and continuous water level measurements for 62 days during the summer of 2016, which is a long record for in-situ supraglacial stream measurements. The manuscript touches on several topics including surface energy budget, the timing of daily maximum discharge, and hydraulic parameterization. The writing is clear and concise. The overall flow is reasonable and easy to follow. However, this manuscript would benefit greatly from more in-depth analyses. Please see my detailed comments below.

1, What's new? I do not mean there is no new information here. However, the take-home messages do not strike me as new knowledge. For example, the conclusion of
the surface energy balance that shortwave radiation dominates the seasonal and diurnal variabilities of surface melt (and thus runoffs) while longwave radiation and sensible heat contribute more than usual to peak melt events has already been made clear by several previous studies cited by this manuscript. The conclusion of the hydraulic parameter analysis concurs with that of Gleason et al. (2016): hydraulic geometry parameters cannot be generalized. The analysis of the timing of daily maximum discharge was left without a firm explanation. A lack of a clear statement of your novelty would make a reader wonder: what questions can your measurements answer but others cannot? Or what are the questions you intend to answer before starting the measurements? What new knowledge did you gain from presenting these measurements?

2, Broader impacts From reading this manuscript, I can tell what you measured and how but it is not clear why. I know these new measurements have merits on their own. I bet they are essential for certain questions. But without explicitly stating these questions and broader impacts, it is hard for readers to connect. For example, the potential impacts of the timing of peak discharges on the subglacial system and ice dynamics should be within the scope of this manuscript not in future work. This is the very reason you measured and analyzed it. Another example is from the introduction (Line 35-38): "...only a handful of studies using in situ supraglacial stream discharge to characterize current conditions". A few more words here to state the caveats of not using in-situ supraglacial stream discharge to study surface mass loss would emphasize the significance of this manuscript.

3, Title A study on "seasonal variability" usually includes more than one season of data. The seasonal variability in this manuscript is more like daily or inter-diurnal variability.

4, Introduction It would be more beneficial for readers if some of the statements are more specific. For example, in Line 32-34, how does increased surface melt influence ice sheet basal properties and ice dynamics: does surface melt increase or decrease basal melt? Line 49-50, how does timing influence ice sheet velocities?

[Figure]

5, Section 2 Study Area How about moving this section to Section 3 as a sub-section?

6, Surface energy balance Surface energy balance largely determines surface melt, which in turn dominates the runoffs. On this daily timescale and without analysis on refreezing, how does your study differ from those using surface melt instead of runoffs, especially when your conclusions are the same as those studies using surface melt?

7, Line 272-273: "As the season progresses, the time to daily maximum discharge will reflect changes in network storage and transport efficiency." I like this sentence. It helps me understand why the timing of peak discharges is important.

8, Section 4.3: this rapid shift in timing is a new discovery. A solid explanation is warranted. How does the temperature change explain this shift?

9, As for all significance tests, are the p values always = 0.00001 or $\leq$ 0.00001?

10, Line 295, what does it mean when the sum of exponent and the product of coefficients does not equal 1? Are they still valid? If the authors do not interpret their results, the readers may take it the wrong way.

11, Line 24-27, I do not think this is a fair comparison considering the large differences in the temporal and spatial scales.

12, Line 373, should the most accurate way be in-situ measurements? Or the next best, reanalysis?

13, Line 375, "lengthy" → "long-term": Using "lengthy" suggests it is too long.

---

## Referee Comment (RC2) · Anonymous Referee #2 · 28 Dec 2020

The authors use a suite of in-situ supraglacial streamflow observations to detail diurnal and seasonal variability in streamflow at the 660 catchment in southwest Greenland. Their analysis shows surface energy flux drivers of surface melt to shift in importance over the timeseries, and that the timing of daily maximum discharge evolves over the season. The authors give a solid introduction and discussion of the existing observations of supraglacial streamflow, and do a thorough job clearly describing the data sets they collected and how they've applied terrestrial hydrology methods to analyze observations of ice-sheet supraglacial hydrology. The figures are largely clear, with a few suggestions listed in the line-by-line comments below.

Prior to publication, however, the authors should work to improve the clarity of their discussion on the seasonal evolution in the timing of daily maximum discharge. In the abstract, a "changing effective catchment area" is listed as a main driver of changes in peak flow timing, but this driver is not fully communicated or supported in the discussion and conclusion sections. Mechanisms driving the direction of the change in lag between solar noon and the timing of daily maximum discharge (i.e., gradually decreasing from three to one hour over the season, and then increasing back to 3 hours at the end of the observation period) are not clearly discussed. The authors mention changes in the weathering crust may be important for both "expansion and contraction" (L396) of the effective catchment area, but give no directionality to this statement (Does the change to the weathering crust they envisage lead to a decrease or expansion of the effective catchment area? Would this result in a longer or shorter lag time?). A clearer communication of the physical changes to the weathering crust and the effects of these changes on the lag and effective catchment area is needed.

If the authors can strengthen their discussion on lag evolution over the season, address the minor comments below, and tighten up the figures, the paper represents a solid effort in conducting and analyzing field observations of supraglacial streamflow, and will be of interest to TC readers.

Minor comments:

L13: Suggest referring to the 46 discharge measurements as "discrete measurements" throughout the manuscript, with "discrete measurements" being in contrast to "continuous" (5-minute) measurements.

L15: Suggest "of supraglacial discharge that captures both..."

L20: Include the baseline percentage contribution to the energy budget from longwave radiation, sensible heat flux, and latent heat flux, and not just the percentage increase of these contributions (as you have for shortwave radiation in the sentence prior). Otherwise, hard for the reader to understand what a percentage increases in these minor

fluxes means in terms of the total energy flux.

L24: Suggest deleting "throughout the melt season"

L25: Suggest: "how widespread rapid shifts in the timing of peak discharge are across…"

L33: Suggest remove citations to studies on Antarctic supraglacial hydrology (Bell et al., 2017; Kingslake et al., 2017), as these references do not address the subject of the clause: "surface melting also influences ice sheet basal properties".

L39: Suggest "melt season" instead of "melting season" throughout (e.g., L45).

L42: Suggest "often terminate in moulins, wherein meltwater moves through and beneath…"

L56: Did you mean to say "time lag decreases" here? As the network contracts, time lag decreases and discharge decreases. This would go with your findings in the abstract on effective catchment area decreasing resulting in a shorter lag.

L61: Suggest delete "thereby surface runoff"

L64: What about "Lesser drivers are…" instead of "Additional drivers are…".

L72: Two commas not needed around Gleason reference

Figure 1: Plot the stream that presumably runs from the discharge station to the moulin. Also suggest coloring stream by stream order.

L112: Suggest "46 discrete discharge measurements" instead of "occasional".

Figure 2: Include approximate scale bars for all panels. An approximate vertical scale bar would improve reader understanding of Figure 2a. Remember that only a few privileged individuals have seen these streams in the field, and many will not have an innate sense of the vertical or horizontal scale.

Figure 3: Keep all units in meters. All other figures are in meters.

[Figure]

Clean up legends for the standard deviation bounds. Would "95% CI" be a clearer legend name here than "2 x stdev"?

The data in Figure 3a is clipped by the top y-axis limit. Show all data.

Maintain the same aspect ratio and axes limits across panels a and b. As presented with differing axes limits and aspect ratios, the panels cannot be compared by eye.

Are these stream cross-sections for the wetted perimeter? If yes, say so. If not, where is the approximate water line (or range in water line)?

L141: Confusion. Should the reference to Figure 4 go here: "The rating curve (Fig. 4; Q=..., R2=...) was..." instead of after the second sentence in this paragraph? As written, the reference to Figure 4 for "hourly discharge data" does not make sense. Hourly discharge data is in Figure 5a, no?

L143: Suggest include the "Therefore, ..." sentence in the previous paragraph. Make the new paragraph begin at "Four uncertainty estimates..."

L144: Suggest "were calculated as percentage uncertainties (see..."

L144: Suggest "discrete" instead of "individual"

L144–145: Give the uncertainty variable names—UME, URC, Xdm, and Uin(?)—in the first sentence where you introduce the four uncertainty estimates.

L146: Are these 95% CI uncertainties?

Figure 4: In legend, indicate whether the "observations of stage and discharge" plotted here are hourly or 5-minute discharge data.

How many sigma are you plotting with URC? Suggest keep the 95% CI language from Figure 3.

L159: Suggest "we compared coincident depth profiles (Fig. 3) and velocity measurements"

L159: Are these velocity measurements ever shown? Show across flow profiles of stream velocity, which you "measured at each 0.2 m interval at 60% of the depth" alongside the stream cross-section depth profiles in Figure 3 or as a stand-alone new figure.

L162Åň–3: Suggest giving 2 sigma values that will better align with the 95% CI given in L164.

L177: Suggest "discharge Q."

L184: State the elevation of KAN–L in comparison to the study site.

L185: State what variables are used from the KAN–L station.

L190: Suggest: "to calculate surface energy balance components of net. . ."

L194: Give a citation for the Monin-Obukhow similarity theory.

L196: Reference Figure 1.

L198: That's a lot of handheld GPS points!

L199: A change in catchment size was not observed, but how does this align with the change in effective catchment area implicated in the abstract? Include a quantification of the time-varying effective catchment area.

L200: State the uncertainty in meters.

L204: Are these repeat visits all during the melt season? As written, the visits could have occurred in the winter as well.

L218–224: The sentence ordering in this section is confusing. Suggest stepping through the description of the timeseries linearly here.

L219: Suggest "Between the second and third episodes, . . ."

L222: Suggest "melt episodes, which have large. . ."

L223: Suggest "amplitude from 0.64 m3s-1 on 21 July to 0.33 m3s-1 on 13 August."

Figure 5: Make all of these panels the same width. Stack them vertically so the eye can travel vertically down the figure to compare the timeseries, just as you have done in Figure 6.

L228: Does the mean daily discharge go all the way to zero? The lowest value in Figure 6a seems to be ~0.02 m3s-1. Suggest defining/describing what zero means when given as the lower bound here. Suggest keeping the same number of places past the decimal when reporting values in text.

Figure 6: Calculate total heat flux/energy from components in panel b, and then plot total heat flux/energy as the right y-axis on panel a.

Figure 7: Note days that are not "clear sky" with vertical bars. State the proportion cloud cover cut off to designate a day as not being "clear sky".

Figure 8: Suggest "melt season".

L273: How different is the time lag on non clear-sky days?

L278: Suggest "shifts back to initial season conditions of a 3 hour time lag in early August."

L282: Suggest "above the ice surface..."

Figure 9: Include the 2 to 3 clear-sky days between melt episodes 1 and 2 in mid-June on panel a to present the entire 62-day timeseries. Plot the time of daily maximum discharge for non clear-sky days with a different symbol in panel a. Would be interesting to see if the times on non clear-sky days are random, and this would further support your choice of investigating the seasonal evolution of lag only on clear-sky days.

In the panel b legend, do you mean "Daily mean air temperature" ?

L300. Give the 660 catchment size here, so the reader can better compare with the

results from other studies that follow.

L304: Does zero discharge mean a dry stream bed at portions of the day? Observations taken outside of the melt season? Suggest defining/describing what zero means when given as the lower bound in a range.

L307: Is this time local time? Suggest stating at the beginning of the paragraph that all times given will be given in the local time of the study sites.

L309: Quantify and give how many times larger the Holmes (1955) catchment is compared to the 660 catchment.

L316: Suggest "In addition to diurnal variability, . . ."

Table 1: Suggest "660 Catchment (this study)" in the bottom row.

Table 1: Again, what do the zeros in discharge signify for the different studies? A dry stream bed for a portion of a day, or taking observations outside of the melt season?

L341: Suggest new paragraph at "For the 660 catchment,. . .", as you are now switching from comparing the total area of the catchment between studies, to the evolution of the catchment over time at 660.

L343: What does Mernild et al. (2006) attribute the change in lag to, and is this supported by your findings at catchment 660? Is it the change in effective catchment area you mention in the abstract? Add a few sentences on this here.

L348: The last sentence shifts back to effective catchment area (poor vs efficient network) as the reason for a shifting time of daily maximum discharge. But the sentence immediately prior discusses changes to the weathering crust. Suggest working through these points more carefully to clearly describe how a decrease in effective catchment area leads to shorter lag times, and how the weathering crust plays into this decrease in effective catchment area. Add citations for the L346 statements on how changes to the weathering crust under colder conditions would affect lag times. Does Yang et al.

[Figure]

(2018) include weathering crust in their model? State this explicitly for readers who have not read Yang et al. (2018).

Table A2: Are all of these reported at 95% CI, or just the final row? Suggest keep everything at 95% CI and state that in the table caption.

L394: Split these ideas into two sentences: (1) a drop from 3 to 1 hours in lag due to a decrease in effective catchment area; and (2) back to 3 hours due to a sudden drop in air temperature that delays melt from leaving the weathering crust.

L396: "expansion and contraction" Doesn't the increase in lag suggest this is an effective expansion of the stream network? A clearer communication of the physical changes to the weathering crust and the effects of these changes on the lag is needed.

L399: Change "notice" to "observe".

L434: "All uncertainties are expressed at the 95% confidence interval." ← This is the sentence that would be very helpful to have early on in the main text Methods section. In general, a consistent level of reporting for the uncertainties is needed to allow the reader to easily compare uncertainties across measurements.

---

## Referee Comment (RC3) · Anonymous Referee #3 · 29 Dec 2020

Supraglacial stream/river networks route large volumes of surface meltwater on the Greenland Ice Sheet but their discharge remains poorly estimated. This paper presents very valuable discharge measurements and continuous water level measurements. These measurements, to my knowledge, are the first long-term (almost entire summer 2016) observations of supraglacial streams and will significantly improve our understanding in Greenland surface hydrology. Therefore, I suggest this paper will be a very good contribution to TC and recommend it to be published.

I have the following comments. Some of these comments are proposed for further discussion so it is fine if the authors cannot address all the questions. But I suggest

the authors consider including some of these comments in the revised paper to make their paper more broadly interesting.

General comments:

This paper compares stream discharge with surface energy fluxes and reveals short-wave radiation as the primary driver of melting (contributing 50-78 % energy). Besides surface energy fluxes, it will be useful to compare stream discharge with modeled surface runoff because these two variables can be directly compared. The catchment boundary is delineated with very high accuracy so this comparison is possible. I assume the comparison is not conducted because the studied catchment is too small to compare with coarse-resolution RCM simulations. If so, I suggest the authors briefly explain it in the revised manuscript.

Additionally, it is necessary to better explain how proportions of melt energy supplies are obtained from discharge-energy correlation analysis in the Abstract and Results.

Specific comments:

line 11, it is not clear to me what "which" means in this sentence, "surface runoff" or "supraglacial stream networks"?

lines 15-17, I think this is not the main finding of the paper and should be moved backwards.

line 29, three papers are cited so it is not clear where 286+-20 Gt comes from.

line 35, it is not straightforward to understand the importance of supraglacial streams from this sentence. I suggest adding "extensive supraglacial stream networks route large volumes of surface meltwater runoff each summer" or similar descriptions to make the logic easier to follow.

line 39, delete "drainage".

line 43, "or" the ocean?

line 54, compared to smaller catchments with similar surface melt intensity, large catchments imply a longer stream network.

line 58, besides weathering crust, meltwater stored in firn or snow can also influence discharge magnitude and timing.

line 100, what is the purpose to introduce "cryoconite"? it is not included in the following analysis.

Figure 1, use a dot to indicate the study area; will it be useful to show the high-resolution satellite image of the study area? Any crevasses or small moulins identified in the catchment? use different line widths to show streams of order 1, 2, and 3.

line 69, why regional climate models (RCMs) are not used in this study?

Figure 2, is it possible to add scale bar in Figure 2a and 2b?

Figure 3, Thermal erosion cannot be identified because the deepest point changes over time. Is there a stable reference point during field-work period? It is fine if there is not because I understand the main point here is to quantify the uncertainties of channel cross section rather than analyzing thermal erosion.

Figure 4, put legend in the figure to save space.

lines 162, "depth measurement errors can be isolated from incision errors", not clear what this sentence means.

lines 177-179, delete this sentence. Instead, add ack = 1, b + f + m = 1 after equations 2-4.

line 180, although the HOBO weather station is not used, it will be useful to add several sentences to explain why it failed to provide continuous meteorological observations.

line 197, the spatial resolution of panchromatic WorldView-1 image is 0.5 m rather than 1 m. Considering showing this image in Figure 1.

line 198, very impressive to identify catchment boundary in such convincing way. Which date was this work conducted? It will be useful to add the date. Also, it will be helpful to add a DEM-derived catchment boundary for comparison (optional though).

lines 200-201, "We estimate the catchment area to be accurate within 5% given that it was manually identified in the field. However, the precise delineation of the catchment is not relevant to the outcome of the study", what is the purpose of this sentence?

line 206, it is surprising that supraglacial streams remain stable at such low-elevation, fast-flowing area. Any reasons? It will be useful to add ice flow velocities.

Figure 6a, I think it is not necessary to calculate uncertainty of daily mean discharge by averaging hourly discharge. Actually, this is just unit transform rather than uncertainty. Also, I think it is more informative to add daily discharge on Figure 5a and move Figure 6b to Figure 5. By doing this, there will be a much illustrative Figure 5.

Figure 8 is not easy to follow. Particularly, it is not clear what "proportion of melt energy" means.

lines 273, what does "network storage" mean?

lines 293-295, delete this sentence.

lines 379-380, this sentence is not easy to follow.

---

## Short Comment (SC1) · 15 Jan 2021

It's great to see another long record of supraglacial stream discharge from Greenland. In your paper you have reviewed some existing data sets and point out that this new record is of unprecedented length. There is one existing long record that you have missed – admittedly quite hard to find – that is about the same duration (from June to August in 2012), and I thought this should be included in your intro & summary table. It may also be interesting for comparison, as it is from a larger stream at a higher location in the same region of the ice sheet. Full details were provided by Wadham et al. (2016), the information on stream gauging is mostly in the supplement to that paper

(https://bg.copernicus.org/articles/13/6339/2016/bg-13-6339-2016-supplement.pdf).

This record was collected similarly to yours, using stage monitoring and a rating curve from discrete discharge measurements, but we used salt dilutions instead of cross section / water velocity. Measuring cross section in this bigger river would have been difficult (and risky) except during a few cold periods with low flow - as a result, all the cross section measurements would have been biased towards low flow conditions. The salt dilutions seemed to work fine though.

It's interesting that you observed little change in the cross section – although we never measured the cross section profile, we found that a single rating curve was adequate through the whole season, which suggests a consistent cross section profile at our site too.

Reference: Wadham et al. (2016), Biogeosciences, 13, 6339-6352, doi:10.5194/bg-13-6339-2016.

---

## Author Comment (AC1) · 16 Mar 2021

**Response to anonymous Referee #1**

**General Comments:**

This manuscript describes the 46 discharge measurements and continuous water level measurements for 62 days during the summer of 2016, which is a long record for in situ supraglacial stream measurements. The manuscript touches on several topics including surface energy budget, the timing of daily maximum discharge, and hydraulic parameterization. The writing is clear and concise. The overall flow is reasonable and easy to follow. However, this manuscript would benefit greatly from more in-depth analyses. Please see my detailed comments below.

1.  What's new? I do not mean there is no new information here. However, the take home messages do not strike me as new knowledge. For example, the conclusion of the surface energy balance that shortwave radiation dominates the seasonal and diurnal variabilities of surface melt (and thus runoffs) while longwave radiation and sensible heat contribute more than usual to peak melt events has already been made clear by several previous studies cited by this manuscript. The conclusion of the hydraulic parameter analysis concurs with that of Gleason et al. (2016): hydraulic geometry parameters cannot be generalized. The analysis of the timing of daily maximum discharge was left without a firm explanation. A lack of a clear statement of your novelty would make a reader wonder: what questions can your measurements answer but others cannot? Or what are the questions you intend to answer before starting the measurements? What new knowledge did you gain from presenting these measurements?

**Author Reply:**

When we revise this manuscript, we will improve the manuscript to add clear statements about the novelty of our study and a detailed introduction to our measurements answering why we collected this data, how do they differ from current available data and what new knowledge did we gain from analyzing it.

Briefly, our revised manuscript will make these points (the text below was also a response to reviewer 3 comments):

i.   Novelty of our study
    a) While the conclusion about surface energy balance is not new, our study is unique by using direct measurements of runoff. Previous studies used weather stations and model simulations to estimate melting and compare with surface energy components. We will rewrite the manuscript to make these points clear.

b) While timing of daily maximum discharge has been estimated using in-situ discharge in previous studies, those studies only span a few days. In our study, using 62-days long in-situ stream discharge, we show that this timing of daily maximum discharge changes over the melt season due to varying catchment conditions. This long term record allows us to make assessments of supraglacial stream dynamics that are more substantiated than other studies that may be subject to random variability within the system. More explanation about this change and the processes affecting it will be given in the manuscript to make this clear.

c) The goal of performing the hydraulic geometry analysis was to see how the 660 catchment parameters compare with the previous studies and if they fall into a range of parameters from similarly sized supraglacial streams. Our parameters compare well with the parameters from Knighton (1981) and Marston (1983), but do not match with streams from Gleason et al. (2016). Indeed, we reach the same conclusion as Gleason et al. (2016), i.e that hydrologic parameters cannot be generalized over the Greenland ice sheet. However, we find that our study is a contribution given the extremely small number of similar studies. In other words, part of the novelty is that Greenland supraglacial stream flow observations are extremely rare, and long observations like ours are even more rare. Additionally, hydraulic parameters of supraglacial streams have shown to be highly spatially variable and this study furthers Gleason's conclusions by analyzing streams closer to the ice edge.

ii. Research questions behind the collection of these unique measurements. We will clarify the research questions driving our work in the introduction. These questions include:

    a) How does the supraglacial stream discharge vary over the entire melt season?

    b) What are the drivers of this discharge throughout the season?

    c) Given the findings of a model study by Yang et al. (2018) on how timing and amplitude of daily discharge change with stream network size and evolution, what does the timing and amplitude of discharge suggest about network evolution in our study catchment?

    d) How do the hydraulic geometry parameters of the streams in 660 catchment compare with the previous studies?

iii. What new knowledge did we gain from these measurements?

a) Surface energy balance:

We agree with the referee that previous studies have already shown some of our findings about discharge and surface energy balance. Specifically, previous work has shown that that while shortwave radiation highly contributes to the seasonal and diurnal variation of surface melt, longwave radiation and turbulent heat fluxes have an increased contribution during peak melt events (Van den Broeke et al., 2011, Fausto et al. 2016). However, all these studies rely on model simulated melt using local weather station data. In contrast to these studies, we have direct observations of melt water runoff. In other words, what is

new in our study is we specifically use runoff in the streams and explore the drivers influencing the runoff throughout the melt season. We also quantified the dependency of stream discharge over different energy components over different time periods (peak flow days and seasonal average). We will clarify this in the revised manuscript.

b) Timing of daily maximum discharge:

Timing and magnitude of meltwater delivery to the subglacial system can influence subglacial hydraulic pressure and ice velocity. From the previous studies we have seen that this timing varies based on catchment size, however this study adds an additional component by showing that time to peak varies significantly throughout the melt season. This is a novel conclusion and shows that changes in supraglacial network geometry due to increased melt rates and changes in weathering crust hydraulics can influence the conveyance of water to the subglacial environment. In addition, a time series of water level in the weathering crust will be added to the analysis that shows a constant decrease in water stored in the weathering crust coinciding with the decrease in lag between timing of daily maximum discharge and solar noon. Less water stored in the weathering crust could lead to an earlier daily maximum discharge as the melt season progresses. We will clarify the discussion about this novel result in the revised manuscript.

iv.   Importance of these measurements:

We will clarify that our dataset fills a huge gap among geoscience dataset. There is only a very small number of similar datasets of Greenland ice sheet supraglacial stream flow data, and only one other dataset, like ours, which covers the majority of the melt season. In other words, there are almost no direct observations of this glacio-hydrological feature. With our unique data, we provide a perspective on the diurnal and daily variation of supraglacial stream discharge over a melt season. For the first time, we can compare direct observations of melt water losses with energy balance drivers of melt, and explore peak discharge timing.

2.   Broader impacts From reading this manuscript, I can tell what you measured and how but it is not clear why. I know these new measurements have merits on their own. I bet they are essential for certain questions. But without explicitly stating these questions and broader impacts, it is hard for readers to connect. For example, the potential impacts of the timing of peak discharges on the subglacial system and ice dynamics should be within the scope of this manuscript not in future work. This is the very reason you measured and analyzed it. Another example is from the introduction (Line 35-38): "...only a handful of studies using in situ supraglacial stream discharge to characterize current conditions". A few more words here to state the caveats of not using in-situ supraglacial stream discharge to study surface mass loss would emphasize the significance of this manuscript.

We will revise the manuscript to clarify why we did this work. Also see our response to your previous question where we explicitly state the research questions driving our work. The revised manuscript will include the following points:

a) Why
We will clarify that a motivation to study supraglacial stream discharge is that the stream discharge is the actual amount of meltwater that is delivered to moulins. As mentioned before, previous studies only examine surface melt (estimated with models and weather station data), which are not affected by processes that influence flow and surface storage of water from when it melts to when it is delivered to the moulin. Recent research has shown that there may be significant disagreement between modeled runoff and in-situ stream measurements (Smith et al., 2017) and therefore applying in-situ measurements of streamflow allows for an analysis of the energy balance components without this potential modeling bias.

Another motivation is previous work that shows: a) the weathering crust stores water (Cooper et al., 2018) and b) the storage is proportional to solar radiation as windy and overcast conditions with higher longwave radiation reduce the storage capacity with higher longwave radiation (Takeuchi et al., 2000). We will add a new time series of cryoconite water level observations that shows a decrease in water level coinciding with the decrease in lag between time of daily maximum discharge and solar noon. By having both stream discharge and cryoconite water level observations we can better address the question about variable storage of water within the weathering crust over the melt season.

b) Potential impacts on ice dynamics
We disagree that an analysis of stream discharge and ice dynamics are within the scope of this paper. We hope that we provided better explanations for the motivation and broader impacts of our study as requested by the reviewer (see response above) so it is clear that our paper fills an important gap in our understanding of Greenland ice sheet melt.

c) Caveats of not using in-situ discharge to study mass loss
In the revised manuscript we will add text to explain what additional knowledge is gained by using in-situ supraglacial stream discharge to study mass loss. To summarize, the main gain of using in-situ discharge is that they provide direct measurements of actual meltwater delivery to moulins by providing both volume and timing. The data can also be used to infer information about surface processes, e.g. temporary storage in weather crust and supraglacial stream network dynamics.

3. Title A study on "seasonal variability" usually includes more than one season of data. The seasonal variability in this manuscript is more like daily or inter-diurnal variability.

**Author Reply:**

We agree with the reviewer and are considering changing the title to: "Supraglacial in situ streamflow and drivers over 62 days of 2016 melt season in Southwest Greenland".

4.  Introduction It would be more beneficial for readers if some of the statements are more specific. For example, in Line 32-34, how does increased surface melt influence ice sheet basal properties and ice dynamics: does surface melt increase or decrease basal melt? Line 49-50, how does timing influence ice sheet velocities?

**Author Reply:**

We will rework the manuscript and include specific information and remove vague statements. We will rewrite the specific examples mentioned by the reviewer and revise other similar statements

5.  Section 2 Study Area How about moving this section to Section 3 as a sub-section?

**Author Reply:**

Our preference is to keep the study area description in its own separate section as is common for many scientific manuscripts. However, we can move the study area section to the methods section if the reviewers and editors feel strongly about this.

6.  Surface energy balance largely determines surface melt, which in turn dominates the runoffs. On this daily timescale and without analysis on refreezing, how does your study differ from those using surface melt instead of runoffs, especially when your conclusions are the same as those studies using surface melt?

**Author Reply:**

We will rewrite the paper to clarify that in agreement with previous work we find that the SEB budget explained the majority of the moulin discharge. However, we also find that evidence of catchment processes are influencing the stream discharge. Most important is probably our finding that the catchment modulates the delivery of surface melt to the moulin so that the timing of peak discharge changes over the season. As far as we know, our documentation of a changing time to peak over the season has not been shown with in-situ data for Greenland ice sheet streams prior to our work.

7.  Line 272-273: "As the season progresses, the time to daily maximum discharge will reflect changes in network storage and transport efficiency." I like this sentence. It helps me understand why the timing of peak discharges is important.

**Author Reply:**

Thank you for the comment on this sentence.

    8.   Section 4.3: this rapid shift in timing is a new discovery. A solid explanation is warranted. How does the temperature change explain this shift?

**Author Reply:**

We appreciate the request for more explanation for the sudden shift in time of peak discharge around early August.

With respect to how a temperature shift explains the rapid shift in peak discharge timing: With the night-time air temperatures close to 1 °C, the skin temperatures could be below freezing. Thus, we suspect that meltwater delivery to the channels are slowed down due to freezing water. With below freezing temperatures, there is likely a change in the manning's n coefficient as frozen ice features pose an impedance to flow, in turn lowering the streams conveyance.

In the revised manuscript, we plan to devote more time to discuss these findings. Unfortunately, we lack many of the auxiliary data that could help us explain the change in the sudden peak. However, we will do a better job to discuss what those auxiliary data could be, i.e. precision stream temperature measurements and skin surface temperatures. We will also do a better job in explaining conceptual models for what could explain the change in peak timing. For example, below freezing surface temperatures could slow down flow, and a switch from channelized to overland flow after the peak melt event would also slow down the flow. Unfortunately, as mentioned, we don't have the observations to prove which conceptual model is dominating in our catchment, but we will discuss what observations are needed to figure it out.

We were recently able to recover a time series of water level observations in cryoconites holes in our catchment that we will add to the paper (see figure below). Our preliminary analysis of this data suggests a drop in the cryoconite water level that occurs around the same time as the change in timing in early August. While this data does not explain the abrupt change in timing that occurs in early august, it is indicative of a connection between the change in timing and reduced water storage in cryoconite holes.

[Figure]

**Fig:** Water level in a cryoconite hole close to the discharge station, measured using a levellogger.

Revised version of Figure 6 will include cryoconite observations of water level and temperature

9. As for all significance tests, are the p values always = 0.00001 or <= 0.00001?

**Author Reply:**
Not all p values are less than or equal to 0.00001 (they are in the range of 0.00001-0.00011). Since they are extremely small, we will simplify the manuscript by just stating that they are ≤ 0.0001 throughout.

10. Line 295, what does it mean when the sum of exponent and the product of coefficients does not equal 1? Are they still valid? If the authors do not interpret their results, the readers may take it the wrong way.

**Author Reply:**
In theory the sum of exponents and the product of coefficients must equal 1, but in practice due to measurement error and when $R^2 < 1.00$, i.e., the power law does not perfectly describe the data, and we can expect deviations from 1. From the previous hydrological studies, we can conclude that if the sum of exponents and the product of coefficients are close to 1 the theory of hydraulic geometry holds. We will clarify this in the revised manuscript.

11. Line 24-27, I do not think this is a fair comparison considering the large differences in the temporal and spatial scales.

**Author Reply:**
We will rewrite this section and clarify that the transferability of our findings may also be affected by scale, i.e. our study catchment is on the smaller scale and also is placed at the lowest

part of the ablation zone.

12. Line 373, should the most accurate way be in-situ measurements? Or the next best, reanalysis?

**Author Reply:**
We will revise this sentence to clarify that regional climate models are the most widely used (instead of accurate) method to estimate surface runoff for the entire Greenland. At this time, it is not logistically possible to estimate all of Greenland runoff in situ methods.

13. Line 375, "lengthy"! "long-term": Using "lengthy" suggests it is too long.

**Author Reply:**
As requested, we will replace "lengthy" with "long-term"

---

## Author Comment (AC2) · 16 Mar 2021

Response to anonymous referee #2

**General Comments:**

The authors use a suite of in-situ supraglacial streamflow observations to detail diurnal and seasonal variability in streamflow at the 660 catchment in southwest Greenland. Their analysis shows surface energy flux drivers of surface melt to shift in importance over the timeseries, and that the timing of daily maximum discharge evolves over the season. The authors give a solid introduction and discussion of the existing observations of supraglacial streamflow, and do a thorough job clearly describing the data sets they collected and how they've applied terrestrial hydrology methods to analyze observations of ice-sheet supraglacial hydrology. The figures are largely clear, with a few suggestions listed in the line-by-line comments below.

Prior to publication, however, the authors should work to improve the clarity of their discussion on the seasonal evolution in the timing of daily maximum discharge. In the abstract, a "changing effective catchment area" is listed as a main driver of changes in peak flow timing, but this driver is not fully communicated or supported in the discussion and conclusion sections. Mechanisms driving the direction of the change in lag between solar noon and the timing of daily maximum discharge (i.e., gradually decreasing from three to one hour over the season, and then increasing back to 3 hours at the end of the observation period) are not clearly discussed. The authors mention changes in the weathering crust may be important for both "expansion and contraction" (L396) of the effective catchment area, but give no directionality to this statement (Does the change to the weathering crust they envisage lead to a decrease or expansion of the effective catchment area? Would this result in a longer or shorter lag time?). A clearer communication of the physical changes to the weathering crust and the effects of these changes on the lag and effective catchment area is needed. If the authors can strengthen their discussion on lag evolution over the season, address the minor comments below, and tighten up the figures, the paper represents a solid effort in conducting and analyzing field observations of supraglacial streamflow, and will be of interest to TC readers.

In the revised manuscript, we will improve and clarify the discussion on the seasonal change in the timing of daily maximum discharge.
This will include a discussion about what auxiliary observations that would help explain the rapid shift in maximum daily discharge in early August. For example, observations of skin temperature and high precision stream temperature could reveal that the lowering of air temperature in early August also resulted in freezing streams. With below freezing temperatures, there is likely a change in the manning's n coefficient as frozen ice features pose an impedance to flow, in turn lowering the streams conveyance.

We will also add a new analysis to the manuscript that will provide some more context to the change in peak timing. New analysis includes a time series of water levels and temperature in a cryoconite hole (in the weathering crust). Our preliminary analysis suggests that the water level drops in the cryoconite hole over the span of ~20 days. The water level drop in the cryoconite hole shows a sudden drop in early August coinciding with the change in peak daily discharge timing. This suggests that the change in peak timing is linked to a reduction of surface water storage in cryoconite holes. We hypothesize that the drop in surface water storage and freezing overland flow melt due to colder temperatures could result in change in peak timing. While we lack the surface temperature observations to test this hypothesis, we will improve our writing and explain what observations are needed to test hypotheses about what is governing the change in peak timing.

**Minor comments:**

1. L13: Suggest referring to the 46 discharge measurements as "discrete measurements" throughout the manuscript, with "discrete measurements" being in contrast to "continuous" (5-minute) measurements.

**Author reply:**

Thank you for the suggestion, we agree that referring them to discrete measurements makes it clear. We will make this change in the revised manuscript.

2. L15: Suggest "of supraglacial discharge that captures both. . ."

**Author reply:**

We accept the suggestion and will add it into the revised manuscript.

3. L20: Include the baseline percentage contribution to the energy budget from longwave radiation, sensible heat flux, and latent heat flux, and not just the percentage increase of these contributions (as you have for shortwave radiation in the sentence prior). Otherwise, it is hard for the reader to understand what a percentage increase in these minor fluxes means in terms of the total energy flux.

**Author reply:**

We accept the suggestion and will include the baseline percentage contributions from longwave radiation, sensible heat flux, and latent heat flux in the revised manuscript.

4. L24: Suggest deleting "throughout the melt season"

**Author reply:**

We accept the suggestion and will delete the text from the revised manuscript.

5.  L25: Suggest: "how widespread rapid shifts in the timing of peak discharge are across. ."

**Author reply:**
We accept the suggestion and will edit the text in the revised manuscript.

6.  L33: Suggest remove citations to studies on Antarctic supraglacial hydrology (Bell et al., 2017; Kingslake et al., 2017), as these references do not address the subject of the clause: "surface melting also influences ice sheet basal properties".

**Author reply:**
We accept the suggestion and will remove both the citations in the revised manuscript.

7.  L39: Suggest "melt season" instead of "melt season" throughout (e.g., L45).

**Author reply:**
We accept the suggestion and will edit the text in the revised manuscript.

8.  L42: Suggest "often terminate in moulins, wherein meltwater moves through and beneath. . ."

**Author reply:**
We accept the suggestion and will edit the text in the revised manuscript.

9.  L56: Did you mean to say "time lag decreases" here? As the network contracts, time lag decreases and discharge decreases. This would go with your findings in the abstract on effective catchment area decreasing resulting in a shorter lag.

**Author reply:**
After carefully re-reading the paper by Yang et al., 2018 we only find strong evidence for decreased stream network and decreased discharge amplitude. We will revise the sentence to make this clear.

10. L61: Suggest delete "thereby surface runoff"

**Author reply:**
We accept the suggestion and will delete the text from the revised manuscript.

11. L64: What about "Lesser drivers are. . ." instead of "Additional drivers are. . .".

**Author reply:**
We accept the suggestion and will replace 'additional drivers' with 'lesser drivers'.

12. L72: Two commas not needed around Gleason reference

**Author reply:**
We accept the suggestion and will remove the extra comma in the revised manuscript.

13. Figure 1: Plot the stream that presumably runs from the discharge station to the moulin. Also suggest coloring stream by stream order.

**Author reply:**
Thank you for the suggestion, we will plot the stream from discharge station to the moulin in the revised manuscript. We will also consider the suggestion about coloring stream by stream order.

14. L112: Suggest "46 discrete discharge measurements" instead of "occasional".

**Author reply:**
We accept the suggestion and will edit the text in the revised manuscript.

15. Figure 2: Include approximate scale bars for all panels. An approximate vertical scale bar would improve reader understanding of Figure 2a. Remember that only a few privileged individuals have seen these streams in the field, and many will not have an innate sense of the vertical or horizontal scale.

**Author reply:**
We appreciate the suggestion and will include vertical and horizontal scale bars for each panel.

16. Figure 3: Keep all units in meters. All other figures are in meters. Clean up legends for the standard deviation bounds. Would "95% CI" be a clearer legend name here than "2 x stdev"? The data in Figure 3a is clipped by the top y-axis limit. Show all data. Maintain the same aspect ratio and axes limits across panels a and b. As presented with differing axes limits and aspect ratios, the panels cannot be compared by eye. Are these stream cross-sections for the wetted perimeter? If yes, say so. If not, where is the approximate water line (or range in water line)?

**Author reply:**

We appreciate and accept all the suggestions and will make appropriate changes in the revised manuscript. Yes, these cross-sections are for the wetted perimeter. We will mention this in the text and figure captions in the revised manuscript.

17. L141: Confusion. Should the reference to Figure 4 go here: "The rating curve (Fig. 4; Q=. . ., R2=. . .) was. . ." instead of after the second sentence in this paragraph? As written, the reference to Figure 4 for "hourly discharge data" does not make sense. Hourly discharge data is in Figure 5a, no?

**Author reply:**
Yes, it was a mistake. The reference for Figure 4 should go after the first sentence in the paragraph. Hourly discharge data is shown in Figure 5a. We will make this correction in the revised manuscript.

18. L143: Suggest include the "Therefore, . . ." sentence in the previous paragraph. Make the new paragraph begin at "Four uncertainty estimates. . ."

**Author reply:**
We accept the suggestion and will shift the sentence to the previous paragraph.

19. L144: Suggest "were calculated as percentage uncertainties (see. . ."

**Author reply:**
We accept the suggestion and will edit the text in the revised manuscript.

20. L144: Suggest "discrete" instead of "individual"

**Author reply:**
We accept the suggestion and will edit the text in the revised manuscript.

21. L144–145: Give the uncertainty variable namesâ˘TUME, URC, Xdm, and Uin(?)⢠A˘Tin˘ the first sentence where you introduce the four uncertainty estimates.

**Author reply:**
We accept the suggestion and will mention the variable names where we introduce the uncertainties.

22. L146: Are these 95% CI uncertainties?

**Author reply:**

Yes, these are 95% CI uncertainties (details explained in Appendix A). We will mention this in the text in the revised manuscript.

23. Figure 4: In legend, indicate whether the "observations of stage and discharge" plotted here are hourly or 5-minute discharge data. How many sigma are you plotting with URC? Suggest keep the 95% CI language from Figure 3.

**Author reply:**
We accept the suggestion and will indicate that the observations of stage are plotted from 5-min stage data and discharge from discrete discharge measurements. The shaded area plotting URC covers 2 sigma (it is mentioned in the text). We will make it clear in the figure caption as well in the revised manuscript. We will also include 95% CI information here in the caption.

24. L159: Suggest "we compared coincident depth profiles (Fig. 3) and velocity measurements"

**Author reply:**
We accept the suggestion and will edit the text in the revised manuscript.

25. L159: Are these velocity measurements ever shown? Show across flow profiles of stream velocity, which you "measured at each 0.2 m interval at 60% of the depth" alongside the stream cross-section depth profiles in Figure 3 or as a stand-alone new figure.

**Author reply:**
The primary purpose of the velocity is to calculate discharge. We choose not to visualize the across flow profiles of stream velocity because it does not directly add to the science questions we are investigating. We are happy to add it in case we are ignoring some process that the reviewer thinks we should consider, please let us know. However, the dataset includes detailed information about the velocity cross sections and can be used by anyone interested in the same.

26. L162Ân–3: Suggest giving 2 sigma values that will better align with the 95% CI given ˇ in L164.

**Author reply:**
We accept the suggestion and will add 2 sigma values for uncertainty due to incision in hourly and daily profiles.

27. L177: Suggest "discharge Q."

**Author reply:**

We accept the suggestion and will edit the text in the revised manuscript.

28. L184: State the elevation of KAN–L in comparison to the study site.

**Author reply:**
We accept the suggestion and will state the elevation of KAN-L station in the revised manuscript.

29. L185: State what variables are used from the KAN–L station.

**Author reply:**
We have only used energy components from KAN-L station in our analysis. We accept the suggestion and will state the variables used from the KAN-L station in the revised manuscript.

30. L190: Suggest: "to calculate surface energy balance components of net. . ."

**Author reply:**
We accept the suggestion and will edit the text in the revised manuscript.

31. L194: Give a citation for the Monin-Obukhov similarity theory.

**Author reply:**
We accept the suggestion and will add a citation for the Monin-Obukhov similarity theory in the revised manuscript.

32. L196: Reference Figure 1.

**Author reply:**
We accept the suggestion and will add the reference for Figure 1 in the revised manuscript.

33. L198: That's a lot of handheld GPS points!

**Author reply:**
Yes, we had set the GPS to record location for every second and walked along the catchment divide.

34. L199: A change in catchment size was not observed, but how does this align with the change in effective catchment area implicated in the abstract? Include a quantification of the time-varying effective catchment area.

**Author reply:**

Effective catchment area is the area that is connected through the stream network that efficiently transports meltwater generated in that area through the network and it varies with change in size of the stream network. As the stream network expands, the area from which it can transport the meltwater is increased and that is what we referred to as the effective catchment area. We plan to change the narrative from effective catchment area to size of the stream network (as the previous explanation leads to more confusion). We will explain this better in the revised manuscript to avoid confusion.

35. L200: State the uncertainty in meters.

**Author reply:**

We accept the suggestion and will add the uncertainty in meters in the revised manuscript.

36. L204: Are these repeat visits all during the melt season? As written, the visits could have occurred in the winter as well.

**Author reply:**

Yes, all these visits were during the melt season. We will clarify this in the revised manuscript.

37. L218–224: The sentence ordering in this section is confusing. Suggest stepping through the description of the timeseries linearly here.

**Author reply:**

We accept the suggestion and will rewrite the paragraph to avoid confusion in the timeseries of the events.

38. L219: Suggest "Between the second and third episodes, . . ."

**Author reply:**

We accept the suggestion and will edit the text in the revised manuscript.

39. L222: Suggest "melt episodes, which have large. . ."

**Author reply:**

We accept the suggestion and will edit the text in the revised manuscript.

40. L223: Suggest "amplitude from 0.64 m3s-1 on 21 July to 0.33 m3s-1 on 13 August."

**Author reply:**

We accept the suggestion and will edit the text in the revised manuscript.

41. Figure 5: Make all of these panels the same width. Stack them vertically so the eye can travel vertically down the figure to compare the timeseries, just as you have done in Figure 6.

**Author reply:**

We accept the suggestion and will edit the figure to align the panels vertically in the revised manuscript.

42. L228: Does the mean daily discharge go all the way to zero? The lowest value in Figure 6a seems to be ~0.02 m3s-1. Suggest defining/describing what zero means when given as the lower bound here. Suggest keeping the same number of places past the decimal when reporting values in text.

**Author reply:**

We appreciate and accept the suggestion. Yes, the lower bound is not exactly zero in daily discharge. We will add the exact lower bound in daily discharge with the same number of digits past decimal as the higher bound.

43. Figure 6: Calculate total heat flux/energy from components in panel b, and then plot total heat flux/energy as the right y-axis on panel a.

**Author reply:**

The point of this figure is to show the radiation components and their variation over the melt season. We choose not to include the total heat flux because it does not add to our analysis and only adds too much information in the figure.

44. Figure 7: Note days that are not "clear sky" with vertical bars. State the proportion cloud cover cut off to designate a day as not being "clear sky".

**Author reply:**

As requested, we will add a shaded region in figure 7b that indicates times with large cloud cover and state a cloud cover threshold (e.g. 0.6).

45. Figure 8: Suggest "melt season".

**Author reply:**

We accept the suggestion and will edit the text in the revised manuscript.

46. L273: How different is the time lag on non clear-sky days?

**Author reply:**
As requested, we will add text to this paragraph to describe the differences between the time lag on clear-sky and cloudy days. The point we will make is that depending on the cloud cover and the time at which clouds disrupt the incoming solar radiation, we either had an earlier peak or a delayed peak in diurnal stream discharge over the day. For example, days with cloud cover for longer than 3-4 hours during the middle of the day (10:00 - 16:00 local time), have a peak discharge earlier in the day around noon to 13:00. On the other hand, days with shorter periods of cloud cover around mid-day, had peak discharge later in the day.

47. L278: Suggest "shifts back to initial season conditions of a 3 hour time lag in early August."

**Author reply:**
We accept the suggestion and will edit the text in the revised manuscript.

48. L282: Suggest "above the ice surface. . ."

**Author reply:**
We accept the suggestion and will edit the text in the revised manuscript.

49. Figure 9: Include the 2 to 3 clear-sky days between melt episodes 1 and 2 in mid-June on panel a to present the entire 62-day timeseries. Plot the time of daily maximum discharge for non clear-sky days with a different symbol in panel a. Would be interesting to see if the times on non clear-sky days are random, and this would further support your choice of investigating the seasonal evolution of lag only on clear-sky days. In the panel b legend, do you mean "Daily mean air temperature" ?

**Author reply:**
We will add an analysis of the time lag on cloud days to the text in this paragraph. This text will present these results: The $R^2$ for trend in peak timing in June and July for clear sky days is 0.79 and statistically significant ($p<0.01$). In contrast, the $R^2$ for trend in peak timing in June and July for all days, cloudy and clear sky, is only 0.15 and not statistically significant. We will add a figure that shows the peak timing of cloudy days to the supplemental material.

In the legend, the first time series is daily mean air temperature. We will clarify this in the revised manuscript.

50. L300. Give the 660 catchment size here, so the reader can better compare with the results from other studies that follow.

**Author reply:**
We accept the suggestion and will add the catchment size here in the revised manuscript.

51. L304: Does zero discharge mean a dry stream bed at portions of the day? Observations taken outside of the melt season? Suggest defining/describing what zero means when given as the lower bound in a range.

**Author reply:**
We will clarify that our range shows the minimum value and the maximum value reported by the authors of the paper. We also took a closer look at the paper by Marston et al. (1983) and found that the minimum value is only shown in a figure and it is very difficult to determine the exact minimum value. However, we can say that the number is close to zero (but not completely zero) and we will revise the text accordingly.

52. L307: Is this time local time? Suggest stating at the beginning of the paragraph that all times given will be given in the local time of the study sites.

**Author reply:**
We accept the suggestion and will state that all the times mentioned here are local time in the revised manuscript.

53. L309: Quantify and give how many times larger the Holmes (1955) catchment is compared to the 660 catchment.

**Author reply:**
We accept the suggestion and will edit the text in the revised manuscript.

54. L316: Suggest "In addition to diurnal variability, . . ."

**Author reply:**
We accept the suggestion and will edit the text in the revised manuscript.

55. Table 1: Suggest "660 Catchment (this study)" in the bottom row.

**Author reply:**

We accept the suggestion and will edit the text in the revised manuscript.

56. Table 1: Again, what do the zeros in discharge signify for the different studies? A dry stream bed for a portion of a day, or taking observations outside of the melt season?

**Author reply:**

We will explain in the table caption that all data presented in the table are from observations made during the melt season. In other words, the zero value represents a dry stream bed in the melt season.

57. L341: Suggest new paragraph at "For the 660 catchment,. . .", as you are now switching from comparing the total area of the catchment between studies, to the evolution of the catchment over time at 660.

**Author reply:**

We accept the suggestion and will edit the text in the revised manuscript.

58. L343: What does Mernild et al. (2006) attribute the change in lag to, and is this supported by your findings at catchment 660? Is it the change in effective catchment area you mention in the abstract? Add a few sentences on this here.

**Author reply:**

Mernild et al. (2006) attributed the change in lag to change in weathering crust structure. We also see such a change in weathering crust, possibly leading to change in lag time. However, they do not talk about change in the effective catchment area/size of stream network. We will make this concept of change in size of stream network clear in the revised manuscript and also write about the findings from Mernild et al. (2006), which we also concur.

59. L348: The last sentence shifts back to the effective catchment area (poor vs efficient network) as the reason for a shifting time of daily maximum discharge. But the sentence immediately prior discusses changes to the weathering crust. Suggest working through these points more carefully to clearly describe how a decrease in effective catchment area leads to shorter lag times, and how the weathering crust plays into this decrease in effective catchment area. Add citations for the L346 statements on how changes to the weathering crust under colder conditions would affect lag times. Does Yang et al. (2018) include weathering crust in their model? State this explicitly for readers who have not read Yang et al. (2018).

**Author reply:**

We accept all the suggestions and will make a clear discussion about the influence of changes in effective catchment area and weathering crust on the change in lag time. We will include citations as suggested in L346.

Yang et al. (2018) do not include weathering crust explicitly in their model, but consider flow patterns which mimic the meltwater flow through the weathering crust. We will explain this in the revised manuscript and make a clear discussion.

60. Table A2: Are all of these reported at 95% CI, or just the final row? Suggest keep everything at 95% CI and state that in the table caption.

**Author reply:**
The individual uncertainties are empirical estimations from the WMO (2010). Final uncertainty estimate from measurement error is calculated with 95% confidence. We will clarify this in the table caption.

61. L394: Split these ideas into two sentences: (1) a drop from 3 to 1 hours in lag due to a decrease in effective catchment area; and (2) back to 3 hours due to a sudden drop in air temperature that delays melt from leaving the weathering crust.

**Author reply:**
We accept the suggestion and will edit the text in the revised manuscript.

62. L396: "expansion and contraction" Doesn't the increase in lag suggest this is an effective expansion of the stream network? A clearer communication of the physical changes to the weathering crust and the effects of these changes on the lag is needed.

**Author reply:**
When the stream network is expanded, the channel flow is reaching a larger area and can transport the meltwater from farther corners of the catchment. However, when the network did not cover channel flow in that area, meltwater generated flows through the weathering crust and overland, causing a slower routing to the nearest stream. Therefore, an efficient network is a longer/expanded stream network which can transport most of the melt generated at a faster rate than overland and subsurface flow. We will rewrite the discussion and conclusions to make this clear in the revised manuscript.

63. L399: Change "notice" to "observe".

**Author reply:**
We accept the suggestion and will edit the text in the revised manuscript.

64. L434: "All uncertainties are expressed at the 95% confidence interval." ← This is the sentence that would be very helpful to have early on in the main text Methods section. In general, a consistent level of reporting for the uncertainties is needed to allow the reader to easily compare uncertainties across measurements.

**Author reply:**
We appreciate and accept the suggestion and will add the text in the methods section in the revised manuscript.

---

## Author Comment (AC3) · 16 Mar 2021

Response to anonymous referee #3

**General comments:**

Supraglacial stream/river networks route large volumes of surface meltwater on the Greenland Ice Sheet but their discharge remains poorly estimated. This paper presents very valuable discharge measurements and continuous water level measurements. These measurements, to my knowledge, are the first long-term (almost entire summer 2016) observations of supraglacial streams and will significantly improve our understanding in Greenland surface hydrology. Therefore, I suggest this paper will be a very good contribution to TC and recommend it to be published. I have the following comments. Some of these comments are proposed for further discussion so it is fine if the authors cannot address all the questions. But I suggest the authors consider including some of these comments in the revised paper to make their paper more broadly interesting.

This paper compares stream discharge with surface energy fluxes and reveals shortwave radiation as the primary driver of melting (contributing 50-78 % energy). Besides surface energy fluxes, it will be useful to compare stream discharge with modeled surface runoff because these two variables can be directly compared. The catchment boundary is delineated with very high accuracy so this comparison is possible. I assume the comparison is not conducted because the studied catchment is too small to compare with coarse-resolution RCM simulations. If so, I suggest the authors briefly explain it in the revised manuscript.
Additionally, it is necessary to better explain how proportions of melt energy supplies are obtained from discharge-energy correlation analysis in the Abstract and Results.

**Author reply:**
We appreciate the suggestion about comparing with modeled runoff from Regional Climate Models. However, the objective of this manuscript is to highlight the novel data set of stream discharge and its drivers over the span of 62-days of the melt season. We are going to revise the manuscript to make our objectives and science questions more clear (see list below, these comments are also included in response to reviewer 1). We think this paper stands on its own, and believe that a model comparison study would require a separate manuscript to do a rigorous model and observation comparison that would provide new insights. Indeed, as the reviewer points out, the RCM model resolution is large compared to this catchment, which requires a more involved study than we think should be added to this manuscript.

Novelty of our study
  a) While the conclusion about surface energy balance is not new, our study is unique by using direct measurements of runoff. Previous studies used weather stations and model

simulations to estimate melting to compare with surface energy components. We will rewrite the manuscript to make these points clearer.

b)  While timing of daily maximum discharge has been estimated using in-situ discharge in previous studies, those studies only span a few days. In our study, using 62-days long in-situ stream discharge, we show that this timing of daily maximum discharge changes over the melt season due to varying catchment conditions. This long term record allows us to make assessments of supraglacial stream dynamics that are more substantiated than other studies that may be subject to random variability within the system. More explanation about this change and the processes affecting it will be given in the manuscript to make this clear.

c)  The goal of performing the hydraulic geometry analysis was to see how the 660 catchment parameters compare with the previous studies and if they fall into a range of parameters from similarly sized supraglacial streams. Our parameters compare well with the parameters from Knighton (1981) and Marston (1983), but do not match with streams from Gleason et al. (2016). Indeed, we reach the same conclusion as Gleason et al. (2016), i.e that hydrologic parameters cannot be generalized over the Greenland ice sheet. However, we find that our study is a contribution given the extremely small number of similar studies. In other words, part of the novelty is that Greenland supraglacial stream flow observations are extremely rare, and long observations like ours are even more rare. Additionally, hydraulic parameters of supraglacial streams have shown to be highly spatially variable and this study furthers Gleason's conclusions by analyzing streams closer to the ice edge.

Research questions behind the collection of these unique measurements. We will clarify the research questions driving our work in the introduction. These questions include:

a)  How does the supraglacial stream discharge vary over the entire melt season?
b)  What are the drivers of this discharge throughout the season?
c)   Given the findings of a model study by Yang et al. (2018) on how timing and amplitude of daily discharge change with stream network size and evolution, what does the timing and amplitude of discharge suggest about network evolution in our study catchment?
d)  How do the hydraulic geometry parameters of the streams in 660 catchment compare with the previous studies?

We actually did not use the discharge-energy correlation analysis. While the proportion of energy components are shown as a ratio to total melt energy. However, their contribution is shown as a percentage by normalizing the proportions from 0-1 and estimating their change after normalization. We will explain this better in the revised manuscript.

**Specific comments:**
1. line 11, it is not clear to me what "which" means in this sentence, "surface runoff" or "supraglacial stream networks"?

**Author reply:**
Here, 'which' meant the evacuation of meltwater. We will rewrite the sentence to avoid misunderstanding.

2. lines 15-17, I think this is not the main finding of the paper and should be moved backwards.

**Author reply:**
We agree that this is not the main finding of the manuscript, and we will de-emphasize it in the abstract.

3. line 29, three papers are cited so it is not clear where 286+-20 Gt comes from.

**Author reply:**
We will revise the manuscript and only cite the paper from which this number is from (i.e. Mouginot et al, 2019)

4. line 35, it is not straightforward to understand the importance of supraglacial streams from this sentence. I suggest adding "extensive supraglacial stream networks route large volumes of surface meltwater runoff each summer" or similar descriptions to make the logic easier to follow.

**Author reply:**
We appreciate the suggestion and will edit the sentence as suggested in the revised manuscript.

5. line 39, delete "drainage".

**Author reply:**
We accept the suggestion and will edit the text in the revised manuscript.

6. line 43, "or" the ocean?

**Author reply:**
It should be "and the ocean" since most water will actually end up there.

7. line 54, compared to smaller catchments with similar surface melt intensity, large catchments imply a longer stream network.

**Author reply:**
We will consider this sentence instead of the one we currently have.

8. line 58, besides weathering crust, meltwater stored in firn or snow can also influence discharge magnitude and timing.

**Author reply:**
We appreciate the suggestion and will edit the text as suggested in the revised manuscript.

9. line 100, what is the purpose to introduce "cryoconite"? it is not included in the following analysis.

**Author reply:**
We plan to add a new time series of water level and temperature in cryoconite holes in the revised manuscript to explain the shift in time to peak discharge. This introduction to cryoconite holes will be necessary to understand the analysis.

10. Figure 1, use a dot to indicate the study area; will it be useful to show the high resolution satellite image of the study area? Any crevasses or small moulins identified in the catchment? use different line widths to show streams of order 1, 2, and 3.

**Author reply:**
As requested, we will add a dot to indicate the study area. We experimented with different background images when we made the study area figure. We will add a third panel that shows an intermediate map where the reader can see where the study area is relative to land, and most likely we will use a satellite image as a background for that map. However, we found that any background images below the stream network did not add any important information for the study, and also made that panel unnecessarily busy and difficult to interpret.

11. line 69, why regional climate models (RCMs) are not used in this study?

**Author reply:**
We agree that RCMs comparison would be very interesting. However, as explained earlier, the objective of this manuscript is to highlight the novel data set of stream discharge and its drivers over the span of 62-days of the melt season. We think this paper stands on its own, and believe that a model comparison study would require a separate manuscript to do a rigorous model and observation comparison that would provide new insights.

12. Figure 2, is it possible to add scale bars in Figure 2a and 2b?

**Author reply:**
We appreciate the request and will add scale bars in 2a and 2b in the revised manuscript.

13. Figure 3, Thermal erosion cannot be identified because the deepest point changes over time. Is there a stable reference point during the field-work period? It is fine if there is not because I understand the main point here is to quantify the uncertainties of channel cross section rather than analyzing thermal erosion.

**Author reply:**
There was a stable reference point while collecting hourly measurements, with which the cross-sections were aligned to in Figure 3b. We will mention this in the text of the revised manuscript. However, we do not have the same throughout the season.

14. Figure 4, put legend in the figure to save space.

**Author reply:**
We accept the suggestion and will move the legend into the figure in the revised manuscript.

15. lines 162, "depth measurement errors can be isolated from incision errors", not clear what this sentence means.

**Author reply:**
The discrete discharge measurements over a cross-section are susceptible to both measurement and incision errors. However, with an assumption that the incision is small during a 26-hour period, the uncertainty in those hourly measurements are caused due to measurement errors alone. This is what we mean by isolating measurement errors from incision errors. We will rewrite and clarify this in the revised manuscript to avoid confusion.

16. lines 177-179, delete this sentence. Instead, add $ack = 1$, $b + f + m = 1$ after equations 2-4.

**Author reply:**
We will not add the equation because these equations only hold for ideal situations. In theory the sum of exponents and the product of coefficients must equal 1, but in practice due to measurement error and when $R^2 < 1.00$, i.e., the power law does not perfectly describe the data, and we can expect deviations from 1. However, we will add these sentences to explain the same.

17. line 180, although the HOBO weather station is not used, it will be useful to add several sentences to explain why it failed to provide continuous meteorological observations.

**Author reply:**

We have decided to completely remove and discard the HOBO weather station data from this manuscript. We have done additional quality control of this data, and found that we are unable to produce an accurate dataset from the HOBO station. The reasons for the poor data are most likely due to poor installation, for example the station was not installed at a fixed elevation above the surface, the station tilt was at times substantial so that the radiation sensors were far from level. Fortunately, a better quality dataset with the variables we got from the HOBO station could be replaced with data from another AWS station operated by the PROMICE project situated at a similar elevation ~7-8 km away. This is the same station that we used for the energy balance calculations.

18. line 197, the spatial resolution of panchromatic WorldView-1 image is 0.5 m rather than 1 m. Considering showing this image in Figure 1.

**Author reply:**

We appreciate your suggestion and will correct the spatial resolution to 0.5 m in the revised manuscript.

19. line 198, very impressive to identify the catchment boundary in such a convincing way. Which date was this work conducted? It will be useful to add the date. Also, it will be helpful to add a DEM-derived catchment boundary for comparison (optional though).

**Author reply:**

We appreciate the comment. We will include the date of boundary GPS measurements in the revised manuscript. However, we will not include a DEM derived catchment boundary. We actually have tried this, but it is not trivial and does not produce a good catchment due to issues such as small snow bridges and other factors explained in a paper by Kang Yang (Yang et al., 2015).

20. lines 200-201, "We estimate the catchment area to be accurate within 5% given that it was manually identified in the field. However, the precise delineation of the catchment is not relevant to the outcome of the study", what is the purpose of this sentence?

**Author reply:**

We added this sentence because we want to communicate that there is some uncertainty regarding the catchment area if people want to use this delineation for other purposes.

> 21. line 206, it is surprising that supraglacial streams remain stable at such low-elevation, fast-flowing areas. Any reasons? It will be useful to add ice flow velocities.

**Author reply:**
As requested, we will add a figure in the supplementary material that shows the ice flow velocities (e.g. https://nsidc.org/data/NSIDC-0725) and discuss any implications on the results.

> 22. Figure 6a, I think it is not necessary to calculate uncertainty of daily mean discharge by averaging hourly discharge. Actually, this is just unit transform rather than uncertainty. Also, I think it is more informative to add daily discharge on Figure 5a and move Figure 6b to Figure 5. By doing this, there will be a much illustrative Figure 5.

**Author reply:**
We are going to keep hourly discharge data in figure 5, and daily discharge data in figure 6. We plan to change panel 5b and 5c so that they are stacked and have the same x-axis length as panel 5a. We want to show the hourly discharge data. Also, we think adding panel 6b to Figure 5 would overload this figure with information. We want to separate the figure that makes a point about discharge variation (Figure 5) and energy balance variation (Figure 6). Furthermore, the gray shaded regions in both Figure 5 and 6 helps intercomparison of these two figures. However, we can remove the uncertainty envelope in Figure 6a if the reviewers and the editors think it is not important to the analysis.

> 23. Figure 8 is not easy to follow. Particularly, it is not clear what "proportion of melt energy" means.

**Author reply:**
Our intention was to show the contribution of individual energy components towards total total melt energy. However, showing percentage is just confusing with negative net longwave radiation and greater than 100% net shortwave radiation contribution (that balances out the negative net longwave radiation). Therefore, we decided to explain the contribution as a ratio to total melt energy (shows the same result as percentage except this will be in fraction) as the individual component's contribution to total melt energy. We will clarify this in the revised manuscript.

> 24. lines 273, what does "network storage" mean?

**Author reply:**

We will rewrite this sentence to explain the following points: The time to daily maximum discharge is influenced by the catchement's ability to evacuate water. The catchment capacity to evacuate water depends on how much water is transported in the stream channels versus overland on the ice surface. How long and wide are the channels, what are the resistance to flow in the channel and ice surface, how much water is trapped in the weathering crust and so on.

25. lines 293-295, delete this sentence.

**Author reply:**
We disagree that this sentence should be deleted. But we will explain better why our exponents do not add up to one as explained earlier.

26. lines 379-380, this sentence is not easy to follow.

**Author reply:**
We appreciate the comment and will rewrite this sentence so that it is easier to understand.

---

## Author Comment (AC4) · 16 Mar 2021

Response to David Chandler

It's great to see another long record of supraglacial stream discharge from Greenland. In your paper you have reviewed some existing data sets and point out that this new record is of unprecedented length. There is one existing long record that you have missed – admittedly quite hard to find – that is about the same duration (from June to August in 2012), and I thought this should be included in your intro & summary table. It may also be interesting for comparison, as it is from a larger stream at a higher location in the same region of the ice sheet. Full details were provided by Wadham et al. (2016), the information on stream gauging is mostly in the supplement to that paper (https://bg.copernicus.org/articles/13/6339/2016/bg-13-6339-2016-supplement.pdf). This record was collected similarly to yours, using stage monitoring and a rating curve from discrete discharge measurements, but we used salt dilutions instead of cross section / water velocity. Measuring the cross-section in this bigger river would have been difficult (and risky) except during a few cold periods with low flow - as a result, all the cross section measurements would have been biased towards low flow conditions. The salt dilutions seemed to work fine though. It's interesting that you observed little change in the cross section – although we never measured the cross section profile, we found that a single rating curve was adequate through the whole season, which suggests a consistent cross section profile at our site too.
Reference: Wadham et al. (2016), Biogeosciences, 13, 6339-6352, doi:10.5194/bg13-6339-2016.

**Author reply:**
Thank you for the suggestion! We are grateful that you altered us about this data set. We will include Wadham et al. (2016) in the introduction and also discuss the comparison with our results. It is interesting to know that the cross section profile at your site has not changed significantly similar to ours. We will rewrite the manuscript to add this study into our discussion.

---

## Author Response (AR1)

This manuscript was revised according to our response to the reviewers- all of our changes are tracked line-by-line in the submitted documents.

In addition to the reviewer's comments, we added two new analyses to support our hypothesis on why we observe the change in timing of daily maximum discharge through the melt season; (1) time series of water level in the weathering crust (cryoconite hole) and, (2) change in hydraulic geometry parameters through the melt season.

We also changed the method of estimating contribution of energy balance components to melt energy. The initial method was to normalize the proportions of melt energy to one by diving individual proportions by the sum of absolute proportions. However, to avoid confusion, we simplified the method to include both positive and negative proportions instead. For example, the proportion of net longwave radiation was changed from -0.32 to -0.08. Initially these proportions were normalized and expressed in percentage. But in the revised manuscript, they are simply expressed as a change in proportion from -0.32 to -.08, corresponding to a change in contribution of 24% to melt energy.

---

## Author Response (AR2)

We have made the edits as requested by the editor and the reviewer. See a detailed response below. In addition, we have made a few additional minor edits to the manuscript text and figures to clarify the manuscript for the reader. A track changes document is provided for your reference.

**Editor comments**

1) clarify what they mean by "seasonal variability" as only one melt season is discussed in the manuscript (e.g. in L250 as suggested by reviewer #2),

We have rewritten the manuscript where we referred to the seasonal variability in our data, and instead used the phrasing such as variability over the melt season. We have also indicated that we refer to our study period from July to August as the melt season.

2) elaborate on the water-level peak at the end of the cryoconite record (i.e., in Fig. 9c),

We consulted our field notes and discovered that the cryoconite data record included a slug test taking place at 2 pm (local time) on Aug. 13. A slug test is when a large volume of water is injected into a hole to examine hydrological properties. This slug experiments were made for another project and not intended for this study. We have redrawn figure 9c and ended the time series at noon Aug. 13, so that the cryoconite water level variations are only influenced by natural drivers.

3) update the Conclusions to match the Discussion section concerning the explanations of the change in peak time lag.

We have updated the conclusions to match the discussion section concerning the explanations of the change in peak time lag

**Other comments by the editor**

L48-49: The editor suggests: "... study on the export of nitrogen from the Greenland ..."

L85-86: For consistency with L398, the authors could remove the Izeboud et al. (2020) reference in L85.

L88 and L470: Noël et al. (2018).

L385: Add a space between "25-50 km2" and "(40-80 times...".

Figure 1 : Add the location of the cryoconite hole as requested by reviewer #2.

Figure 3: Could the authors align Figs. 3a and 3b on their x-axis (i.e., so that the "-2" at the bottom left in both graphs are aligned)?

Table A1: For variables Ume to Swl, use a capital letters for "Uncertainty" and "Standard" in the "description" column.

Table A2: For variable Um add brackets around "(for a minimum ... 10 verticals)".

For variable Uc add the missing bracket after "ms-1)".

For variable M add the missing bracket after "10-16)".

For variable K add brackets around "(for 95% CI)".

We appreciate and accept all the above suggestions by the editor and made appropriate edits to the manuscript to incorporate them.

**Reviewer comments**

L114: Remove gendered language: "uncrewed" instead of "unmanned."

As requested, we have removed gendered language

L250: Figure 5a does not depict a strong seasonal change in discharge. The lower discharge values in August are pretty much the same as the ones in the late June and early July (those periods outside of the melt episodes). The observations are made within the melt season, and do not show melt onset or cessation.

Where is the cryoconite hole in the study area? Plot on Figure 1.

L339: The last sentence of this paragraph is a repeat of the preceding sentence.

Fig 9: What is going on at the end of the record of cryoconite water-level record in panel c? Why does the water level rapidly increase when daily minimum air temperature dips below freezing?

L431: "...decreasing the storage capacity" Storage capacity of what? The weathering crust? Clarify language for the reader.

L481: Observations do not show seasonal changes in stream discharge (Figure 5).

L483: Where are the seasonal changes in discharge amplitude shown? Outside of the melt events, discharge in August seems pretty much the same as in late June/early July.

L491: The sentence that begins "The change in peak time lag through the melt season…" does not match the explanation of the slow decline in time lag over the melt season given in section 5 (Discussion). In the Conclusion, the process given to explain the declining lag is an expansion or contraction of the stream network (As an aside, shouldn't it be either an expansion or contraction, but not both? The lag declines over the longer time period, so this should be a contraction, no?). In the Discussion, the process given to explain the declining lag is a change in the ratio of channelized to porous flow. Make sure the Conclusion section is updated to reflect changes made in the Discussion section. Better to spell out the two changes independently (i.e., (1) the slow decline in lag and (2) the abrupt change) than to try to meld the interpretations for both of these into the same couple of sentences.

We have followed the reviewer's suggestion to spell out the two changes independently. In the revised document the discussion and conclusion section align.

We appreciate and accept all the above suggestions by the reviewer and made appropriate edits to the manuscript to incorporate them.

---

## Author Response (AR3)

The manuscript was revised according to the suggestions from the Editor. All the changes made were tracked line-by-line in the submitted document. Apart from the suggested comments from the editor, a new reference was cited in the revised manuscript that seemed more appropriate to the discussion in Page 21, line 440.

There is only one issue we would like to bring to the Editor's notice. Not adding 'the' before 'weathering crust' in the entire document seemed more accurate technically. Adding 'the' before 'weathering crust' sounds similar to adding 'the' before 'snow' (like saying meltwater is stored in the snow). We would like to treat 'weathering crust' similar to 'snow' but would be willing to change it if the Editor feels strongly about it. For keeping the text uniform throughout the manuscript, 'the' was removed from all the instances before 'weathering crust' except for when discussing about a specific weathering crust layer.

We are working on getting PANGEA data link and will update as soon as we get it. We would like to request some time to get the link/DOI.